# Efficient Model-Free Exploration in Low-Rank MDPs

**Zakaria Mhammedi**
MIT
mhammedi@mit.edu

**Adam Block**
MIT
ablock@mit.edu

**Dylan J. Foster**
Microsoft Research
dylanfoster@microsoft.com

**Alexander Rakhlin**
MIT
rakhlin@mit.edu

## Abstract

A major challenge in reinforcement learning is to develop practical, sample-efficient algorithms for exploration in high-dimensional domains where generalization and function approximation is required. *Low-Rank Markov Decision Processes*—where transition probabilities admit a low-rank factorization based on an unknown feature embedding—offer a simple, yet expressive framework for RL with function approximation, yet existing algorithms either (1) are computationally intractable, or (2) require restrictive statistical assumptions such as latent variable structure or access to model-based function approximation. In this work, we propose the first provably sample-efficient algorithm for exploration in Low-Rank MDPs that is both computationally efficient and model-free, allowing for general function approximation while requiring no structural assumptions beyond a reachability condition that we show is substantially weaker than that assumed in prior work. Our algorithm, SpanRL, uses the notion of a *barycentric spanner* for the feature embedding as an efficiently computable basis for exploration, performing efficient spanner computation by interleaving representation learning and policy optimization subroutines. Our analysis—which is appealingly simple and modular—carefully combines several techniques, including a new approach to error-tolerant barycentric spanner computation, and a new analysis of a certain minimax representation learning objective found in prior work.

## 1 Introduction

In reinforcement learning and control, many of the most promising application domains require the agent to navigate complex, high-dimensional state and action spaces, where generalization and function approximation is necessary. The last decade has witnessed impressive empirical success in domains where data are abundant [34, 38, 26, 28, 27], but when data are limited, ensuring efficient exploration in large domains is a major research question. For *statistical efficiency*, the foundations have recently begun to take shape, with a line of research providing structural conditions that facilitate sample-efficient exploration, as well as fundamental limits [37, 21, 39, 42, 14, 24, 17, 18]. *Computational efficiency*, however, remains a major challenge: outside of simple settings [7, 23], existing algorithms with provable sample complexity guarantees are computationally inefficient, and typically require solving intractable non-convex optimization problems [21, 11, 24, 10]. The prospect of developing practical algorithms for exploration in high-dimensional state spaces that are both computationally and statistically efficient raises three fundamental questions:

1. What are the right computational primitives for exploration? That is, how can one efficiently represent and compute exploratory policies that allow the learner to explore the state space and gather useful data?

2. How should one leverage function approximation—for example, via representation learning—to discover such primitives in a computationally and statistically efficient fashion?

3. Given answers to the first two questions, how can one efficiently interleave function approximation and exploration to provide provably efficient algorithms?

37th Conference on Neural Information Processing Systems (NeurIPS 2023).

In this paper, we investigate these questions through the *Low-Rank MDP* model [36, 45, 1]. In a Low-Rank MDP, the state space is large and potentially continuous, but the transition probabilities admit an (unknown) low-rank factorization. Concretely, for a finite-horizon Low-Rank MDP with horizon $H$, the transition densities for layer $h \in [H]$ satisfy

$$T_h(x_{h+1} \mid x_h, a_h) = \mu_{h+1}^\star(x_{h+1})^\top \phi_h^\star(x_h, a_h), \tag{1}$$

where $\phi_h^\star(\cdot, \cdot) \in \mathbb{R}^d$ and $\mu_{h+1}^\star(\cdot) \in \mathbb{R}^d$ are state-action and next-state embeddings. The low-rank structure in (1) facilitates tractable exploration: if the embedding $\phi_h^\star$ is known to the learner, one can efficiently learn a near-optimal policy with sample complexity polynomial in the feature dimension $d$, and independent of the size of the state space [23]; in this regard, $\phi_h^\star$ can be thought of as a low-dimensional *representation* that enables sample-efficient RL. Following Agarwal et al. [1], we consider the challenging setting in which both $\phi_h^\star$ and $\mu_{h+1}^\star$ are *unknown* to the learner. This formulation generalizes well-known frameworks such as the *Block MDP* (BMDP) model [12, 32], and necessitates the use of *representation learning*: the agent must learn an embedding that approximates $\phi_h^\star$ as it explores the environment, and must use this learned embedding to drive subsequent exploration. This form of function approximation allows for great flexibility, as $\phi_h^\star$ can be an arbitrary, nonlinear function of the state; in practice, it is common to model $\phi_h^\star$ as a neural net [49].

The Low-Rank MDP is perhaps the simplest MDP structure that demands systematic exploration and nonlinear function approximation while allowing for a continuum of states, yet understanding of *efficient* algorithm design for this model is surprisingly limited. Existing algorithms suffer from at least one of the following drawbacks:

1. Computational intractability [21, 24, 14, 9, 43].

2. Strong modeling assumptions (e.g., ability to model $\mu_{h+1}^\star(\cdot)$, which facilitates application of model-based RL techniques) [1, 40, 10]; in this work, we aim for *model-free* methods that only require learning $\phi_h^\star$.

3. Restrictive structural assumptions (e.g., non-negativity or latent variable structure for the embeddings in (1)) [35, 49].

At the root of these limitations is the complex interplay between exploration and representation learning: the agent must learn a high-quality representation to guide in exploring the state space, but learning such a representation requires gathering diverse and informative data, which is difficult to acquire without having already explored the state space to begin with. Overcoming this challenge—particularly where computational efficiency is concerned—requires (1) representation learning procedures that lead to sufficiently expressive representations for downstream applications, (2) efficient exploration procedures that are robust to errors in learned representations, and 3) understanding the interaction between these procedures, which must be interleaved. In this work, we propose an algorithm that addresses each of these challenges, as detailed below.

**Contributions.** We provide the first provably computationally efficient and model-free algorithm for general Low-Rank MDPs. Our algorithm, SpanRL, uses the notion of a *barycentric spanner* [6] for the embedding $\phi_h^\star$ as an efficiently computable basis for exploration, and combines this with a minimax representation learning approach [35, 49]. SpanRL interleaves exploration with representation learning in a layer-wise fashion, learning a new representation at each layer $h$ using exploratory data gathered at previous layers, then uses this representation to facilitate computation of a collection of exploratory policies (a *policy cover*), which act as an approximate barycentric spanner for the features at layer $h + 1$, ensuring good coverage for subsequent iterations. SpanRL is simple and modular, and its analysis is surprisingly compact given the greater generality compared to prior work [49, 35, 31].

SpanRL can accommodate general-purpose function approximation to learn the representation $\phi^\star$ (e.g., neural nets or other flexible classes) whenever a certain minimax representation learning objective [35, 49] can be solved efficiently for the function class of interest. Compared to efficient algorithms from prior work, SpanRL: (1) is model-free (i.e., only requires access to a function class $\Phi$ capable of modeling $\phi^\star$, and does not need to model $\mu_{h+1}^\star$), and (2) applies to general Low-Rank MDPs, replacing strong additional assumptions such as non-negativity of the feature embeddings (so-called *latent variable* structure) or block structure (see Table 1) with a reachability assumption that we show is substantially weaker than that assumed in prior work (see Appendix H). As a secondary benefit, the algorithm is reward-free. Our analysis carefully combines several new techniques, including (1) an error-tolerant variant of the classical barycentric spanner computation algorithm of Awerbuch

Table 1: Comparison of sample complexity required learn an $\varepsilon$-optimal policy. $\Phi$ denotes the feature class, and $\Upsilon$ denotes an additional feature class capturing model-based function approximation. For approaches that require non-negative (latent variable) structure, $d_{\mathsf{LV}}$ [resp. $\gamma$] denotes the latent variable dimension [resp. the reachability parameter in the latent representation], and for BMDPs, $|\mathcal{S}|$ denotes the size of the latent state space. For SpanRL, $\eta$ denotes the reachability parameter.

| | Comp. efficient | Model-free | General low rank | Sample comp. |
|---|:---:|:---:|:---:|:---:|
| OLIVE [21] (see also [24, 14, 9, 43]) | ✗ | ✓ | ✓ | $\frac{d^3 A H^5 \log|\Phi|}{\varepsilon^2}$ |
| FLAMBE [1] | ✓ | ✗ | ✓[1] | $\frac{d^7 A^9 H^{22} \log(|\Phi\|\Upsilon|)}{\varepsilon^{10}}$ |
| Rep-UCB [40] (see also [10]) | ✓ | ✗ | ✓ | $\frac{d^4 A^2 H^5 \log(|\Phi\|\Upsilon|)}{\varepsilon^2}$ |
| MOFFLE [35][2] | ✓ | ✓ | ✗ Non-negative/ latent variable | $\frac{d_{\mathsf{LV}}^{19} A^{32} H^{19} \log|\Phi|}{(\varepsilon^6 \gamma^3 \wedge \gamma^{11})}$ |
| BRIEE [49] | ✓ | ✓ | ✗ Block MDP | $\frac{|\mathcal{S}|^8 A^{14} H^9 \log|\Phi|}{\varepsilon^4}$ |
| SpanRL (this paper) | ✓ | ✓ | ✓ | $\frac{A^4 d^9 H^4 (d + \log|\Phi|)}{\varepsilon^2 \wedge \eta^2}$ |

and Kleinberg [6], and (2) a new analysis of a minimax representation learning objective introduced in Modi et al. [35], Zhang et al. [49], which shows for the first time that this objective can lead to meaningful guarantees in general Low-Rank MDPs without latent variable structure; this increased generality is meaningful, as we show in Appendix H that there is an exponential separation between our guarantees and those that require such a structure.

**Organization.**  Section 2 formally introduces the Low-Rank MDP model and the online reinforcement learning framework we consider. In Section 3, we highlight challenges faced by previous approaches, introduce our main algorithm, SpanRL, and show how it overcomes these challenges, and then present its main sample complexity guarantee.

**Comparison to ArXiv Version.**  After the initial submission of this work, we developed a substantially improved of the algorithm that removes the reachability assumption at the cost of a larger (but still polynomial) sample complexity guarantee. We have included this algorithm and its analysis in the ArXiv version of this paper [30].

## 2   Problem Setting

### 2.1   Low-Rank MDP Model

We work in an episodic, finite-horizon reinforcement learning framework, where $H \in \mathbb{N}$ denotes the horizon. A *Low-Rank MDP* [36, 45, 1] is a tuple $\mathcal{M} = (\mathcal{X}, \mathcal{A}, (\phi_h^\star)_{h \in [H]}, (\mu_h^\star)_{h \in [H]}, \rho)$ consisting of a *state space* $\mathcal{X}$, *action space* $\mathcal{A}$ with $|\mathcal{A}| = A$, distribution over initial states $\rho \in \Delta(\mathcal{X})$, and mappings $\mu_{h+1}^\star : \mathcal{X} \to \mathbb{R}^d$ and $\phi_h^\star : \mathcal{X} \times \mathcal{A} \to \mathbb{R}^d$.[3] Beginning with $\boldsymbol{x}_1 \sim \rho$, an episode proceeds in $H$ steps, where for each step $h \in [H]$, the state $\boldsymbol{x}_h$ evolves as a function of the agent's action $\boldsymbol{a}_h$ via

$$\boldsymbol{x}_{h+1} \sim T_h(\cdot \mid \boldsymbol{x}_h, \boldsymbol{a}_h),$$

where $T_h$ is a probability transition kernel, which is assumed to factorize based on $\phi_h^\star$ and $\mu_h^\star$. In detail, we assume that there exists a $\sigma$-finite measure $\nu$ on $\mathcal{X}$ such that for all $1 \le h \le H - 1$, and for all $x \in \mathcal{X}$ and $a \in \mathcal{A}$, the function $x' \mapsto \mu_{h+1}^\star(x')^\top \phi_h^\star(x, a)$ is a probability density with respect to $\nu$ (i.e. the function is everywhere non-negative and integrates to 1 under $\nu$). For any $\mathcal{X}' \subseteq \mathcal{X}$, the probability that $\boldsymbol{x}_{h+1} \in \mathcal{X}'$ under $\boldsymbol{x}_{h+1} \sim T_h(\cdot \mid x_h, a_h)$ is then assumed to follow the law

$$T_h(\mathcal{X}' \mid x_h, a_h) = \int_{\mathcal{X}'} \mu_{h+1}^\star(x)^\top \phi_h^\star(x_h, a_h) \mathrm{d}\nu(x). \tag{2}$$

For notational compactness, we assume (following, e.g., Jiang et al. [21]) that the MDP $\mathcal{M}$ is *layered* so that $\mathcal{X} = \mathcal{X}_1 \cup \cdots \cup \mathcal{X}_H$ for $\mathcal{X}_i \cap \mathcal{X}_j = \varnothing$ for all $i \neq j$, where $\mathcal{X}_h \subseteq \mathcal{X}$ is the subset of states in $\mathcal{X}$

---

[1]For the stated sample complexity, FLAMBE requires access to a sampling oracle for the learner model. Without this oracle, the results require additional latent variable structure and a reachability assumption.

[2]We compare to the variant of MOFFLE that uses the same representation learning objective we consider. Other variants have improved sample complexity, but make use of stronger oracles.

[3]We emphasize that neither $\mu_h^\star$ nor $\phi_h^\star$ is known to the agent, in contrast to the linear MDP setting [44, 23].

that are reachable at layer $h \in [H]$. This can be seen to hold without loss of generality (modulo dependence on $H$), by augmenting the state space to include the layer index.

**Remark 2.1** (Comparison to previous formulations). *Our formulation, in which the transition dynamics (2) are stated with respect to a base measure $\nu$, are a rigorous generalization of Low-Rank MDP formulations found in previous works [23, 1], which tend to implicitly assume the state space is countable and avoid rigorously defining integrals. We adopt this more general formulation to emphasize the applicability our results to continuous domains. However, in the special case where state space is countable, choosing $\nu$ as the counting measure yields $T_h(\mathcal{X}' \mid x_h, a_h) = \sum_{x \in \mathcal{X}'} \mu_{h+1}^\star(x)^\top \phi_h^\star(x_h, a_h)$, which is consistent with prior work.*

**Policies and occupancy measures.** We define $\Pi_\mathsf{M} = \{\pi : \mathcal{X} \to \Delta(\mathcal{A})\}$ as the set of all randomized, Markovian policies. For a policy $\pi \in \Pi_\mathsf{M}$, we let $\mathbb{P}^\pi$ denote the law of $(\boldsymbol{x}_1, \boldsymbol{a}_1), \ldots, (\boldsymbol{x}_H, \boldsymbol{a}_H)$ under $\boldsymbol{a}_h \sim \pi(\boldsymbol{x}_h)$, and let $\mathbb{E}^\pi$ denote the corresponding expectation. For any $\mathcal{X}' \subseteq \mathcal{X}_h$, we let $\mathbb{P}_h^\pi[\mathcal{X}'] \coloneqq \mathbb{P}^\pi[\boldsymbol{x}_h \in \mathcal{X}']$ denote the marginal law of $\boldsymbol{x}_h$ under $\pi$. For $x \in \mathcal{X}_h$, we define the *occupancy measure* $d^\pi(x) \coloneqq \frac{\mathrm{d}\mathbb{P}_h^\pi}{\mathrm{d}\nu}(x)$ as the density of $\mathbb{P}_h^\pi$ with respect to $\nu$.

## 2.2 Online Reinforcement Learning and Reward-Free Exploration

We consider a standard *online reinforcement learning* framework where the Low-Rank MDP $\mathcal{M}$ is unknown, and the learning agent interacts with it in *episodes*, where at each episode the agent executes a policy of the form $\pi : \mathcal{X} \to \Delta(\mathcal{A})$ and observes the resulting trajectory $(\boldsymbol{x}_1, \boldsymbol{a}_1), \ldots, (\boldsymbol{x}_H, \boldsymbol{a}_H)$. While the ultimate goal of reinforcement learning is to optimize a policy with respect to a possibly unknown reward function, here we focus on the problem of *reward-free exploration*, which entails learning a collection of policies that almost optimally "covers" the state space, and can be used to efficiently optimize any downstream reward function [12, 33, 15, 31]. To wit, we aim to construct an *policy cover*, a collection of policies that can reach any state with near-optimal probability.

**Definition 2.1** (Policy cover). *For $\alpha \in (0, 1]$, a subset $\Psi \subseteq \Pi_\mathsf{M}$ is an $\alpha$-policy cover for layer $h$ if*

$$\forall x \in \mathcal{X}_h, \quad \max_{\pi \in \Psi} d^\pi(x) \geq \alpha \cdot \max_{\pi' \in \Pi_\mathsf{M}} d^{\pi'}(x). \tag{3}$$

We show (Appendix G) that given access to such a policy cover with constant $\alpha$, it is possible to optimize any downstream reward function with polynomial sample complexity.

**Assumptions.** To facilitate learning a policy cover, we make the following *reachability* assumption.

**Assumption 2.1** ($\eta$-reachability). *For any $h \in [H]$ and $x \in \mathcal{X}_h$, $\max_{\pi \in \Pi_\mathsf{M}} d^\pi(x) \geq \eta \cdot \|\mu_h^\star(x)\|$.*

Reachability is necessary if one aims to build a policy cover that satisfies (3) uniformly for all states; without such a condition, one gives up on covering hard-to-reach states. Some notion of reachability is required in essentially all prior work on efficient model-free algorithms for Low-Rank MDPs [32, 35, 5], and was only very recently removed in the (more restrictive) BMDP setting [31, 49].

**Remark 2.2** (Comparison to other reachability-like assumptions). *Assumption 2.1 generalizes and subsumes all previous reachability-like conditions of which we are aware [33, 46, 1, 35]. Notably, reachability is implied by the notion of* feature coverage *[5] (used in the context of transfer learning in Low-Rank MDPs), which asserts that $\sup_{\pi \in \Pi_\mathsf{M}} \lambda_{\min}(\mathbb{E}^\pi[\phi_h^\star(\boldsymbol{x}_h, \boldsymbol{a}_h)\phi_h^\star(\boldsymbol{x}_h, \boldsymbol{a}_h)^\top]) \geq \eta$, for some $\eta > 0$. It is also implied by* explorability *[46], which is similar to feature coverage, but involves the first moments of $\phi_h^\star$. Our reachability assumption is also weaker than that used in [1, 35] under the* latent variable model*, and generalizes that made for BMDPs [33]. See Appendix H for details, as well as an exponential separation between our assumptions and analogous assumptions in [1, 35].*

Beyond reachability, we assume (following [1, 35]) for normalization that, for all $h \in [H]$ and $(x, a) \in \mathcal{X}_h \times \mathcal{A}$, $\|\phi_h^\star(x, a)\| \leq 1$, and that for all $g : \mathcal{X}_h \to [0, 1]$,

$$\left\| \int_{\mathcal{X}_h} \mu_h^\star(x) g(x) \mathrm{d}\nu(x) \right\| \leq \sqrt{d}. \tag{4}$$

**Function approximation and desiderata.** We do not assume that the true features $(\phi_h^\star)_{h \in [H]}$ or the mappings $(\mu_h^\star)_{h \in [H]}$ are known to the learner. To provide sample-efficient learning guarantees we make use of function approximation as in prior work [3, 35], and assume access to a *feature class* $\Phi \subseteq \{\phi : \mathcal{X} \times \mathcal{A} \to \mathbb{R}^d\}$ that contains $\phi_h^\star$, for $h \in [H-1]$.

**Assumption 2.2** (Realizability). *The feature class $\Phi \subseteq \{\phi : \mathcal{X} \times \mathcal{A} \to \mathbb{R}^d\}$ has $\phi_h^\star \in \Phi$ for all $h \in [H]$. Moreover, for all $\phi \in \Phi$, $x \in \mathcal{X}$, and $a \in \mathcal{A}$, it holds that $\|\phi(x,a)\| \leq 1$.*

The class $\Phi$ may consist of linear functions, neural networks, or other standard models depending on the application, and reflects the learner's prior knowledge of the underlying MDP. We assume that $\Phi$ is finite to simplify presentation, but extension to infinite classes is straightforward, as our results only invoke finiteness through standard uniform convergence arguments. Note that unlike model-based approaches [1, 40, 10, 2], we do not assume access to a class capable of realizing the features $\mu_h^\star$, and our algorithm does not attempt to learn these features; this is why we distinguish our results as *model-free*.

For constant $\alpha$, our goal is to learn an $\alpha$-policy cover using $\mathrm{poly}(d, A, H, \log|\Phi|, \eta^{-1})$ episodes of interaction. This guarantee scales with the dimension $d$ of the feature map and the complexity $\log|\Phi|$ of the feature class but, critically, does not depend on the size of the state space $\mathcal{X}$; by [10], dependence on $H$ and $A = |\mathcal{A}|$ is necessary when $\phi^\star$ is unknown. Given such a guarantee, we show in Appendix G how to optimize any downstream reward function to error $\varepsilon$ with polynomial sample complexity.

**Additional preliminaries.** For any $m, n \in \mathbb{N}$, we denote by $[m..n]$ the integer interval $\{m, \ldots, n\}$. We also let $[n] \coloneqq [1..n]$. For any sequence of objects $o_1, o_2, \ldots$, we define $o_{m:n} \coloneqq (o_i)_{i \in [m..n]}$. A *partial policy* is a policy defined over a contiguous subset of layers $[\ell..r] \subseteq [H]$. We denote by $\Pi_{\mathsf{M}}^{\ell:r} \coloneqq \{\pi : \bigcup_{h=\ell}^r \mathcal{X}_h \to \Delta(\mathcal{A})\}$ the set of all partial policies over layers $\ell$ to $r$; note that $\Pi_{\mathsf{M}} \equiv \Pi_{\mathsf{M}}^{1:H}$. For a policy $\pi \in \Pi_{\mathsf{M}}^{\ell:r}$ and $h \in [\ell..r]$, $\pi(x_h)$ denotes the action distribution for the policy at layer $h$ when $x_h \in \mathcal{X}_h$ is the current state. For $1 \leq t \leq h \leq H$ and any pair of partial policies $\pi \in \Pi_{\mathsf{M}}^{1:t-1}, \pi' \in \Pi_{\mathsf{M}}^{t:h}$, we define $\pi \circ_t \pi' \in \Pi_{\mathsf{M}}^{1:h}$ as the partial policy given by $(\pi \circ_t \pi')(x_\ell) = \pi(x_\ell)$ for all $\ell < t$ and $(\pi \circ_t \pi')(x_\ell) = \pi'(x_\ell)$ for all $\ell \in [t..h]$.

We use the $\boldsymbol{x}_h \sim \pi$ as shorthand to indicate that $\boldsymbol{x}_h$ is drawn from the law $\mathbb{P}^\pi$, and likewise for $(\boldsymbol{x}_h, \boldsymbol{a}_h) \sim \pi$ and so on. For a set of partial policies $\Psi \coloneqq \{\pi^{(i)} : i \in [N]\}$, we define $\mathrm{unif}(\Psi)$ as the random partial policy obtained by sampling $\boldsymbol{i} \sim \mathrm{unif}([N])$ and playing $\pi^{(\boldsymbol{i})}$. We define $\pi_{\mathrm{unif}} \in \Pi_{\mathsf{M}}$ as the random policy that selects actions in $\mathcal{A}$ uniformly at random at each layer. We use $\|\cdot\|$ to denote the Euclidean norm, $\|\cdot\|_\infty$ to denote the supremum norm on functions, and let $\mathcal{B}(r) \subseteq \mathbb{R}^d$ denote the Euclidean ball of radius $r$. We refer to a scalar $c > 0$ as an *absolute constant* to indicate that it is independent of all problem parameters and use $\widetilde{O}(\cdot)$ to denote a bound up to factors polylogarithmic in parameters appearing in the expression.

## 3 SpanRL: Algorithm and Main Results

In this section, we present the SpanRL algorithm. We begin by describing challenges in deriving efficient, model-free algorithms using existing approaches (Section 3.1). We then formally describe SpanRL (Section 3.2) and build intuition as to how it is able to overcome these challenges, and finally state our main sample complexity guarantee (Section 3.3).

### 3.1 Challenges and Related Work

Designing algorithms with provable guarantees in the Low-Rank MDP setting is challenging because of the complicated interplay between representation learning and exploration. Indeed, while there are many efficient algorithms for the so-called *linear MDP* setting where the feature maps $(\phi_h^\star)_{h \in [H]}$ are known (removing the need for representation learning) [23, 47, 4, 41], these approaches do not readily generalize to accommodate unknown features. For Low-Rank MDPs, previous algorithms suffer from at least one of the following three drawbacks: (1) the algorithms are computationally inefficient; (2) the algorithms are model-based; or (3) the algorithms place strong assumptions on the MDP that are unlikely to hold in practice. To motivate the SpanRL algorithm, we briefly survey these results, highlighting several key challenges in avoiding these pitfalls.

Let us first discuss the issue of computational efficiency. While there are a number of algorithms—all based on the principle of *optimism in the face of uncertainty*—that provide tight sample complexity guarantees for Low-Rank MDPs in reward-based [21, 24, 14] and reward-free [9, 43] settings, these algorithms involve intractable optimization problems, and cannot be implemented efficiently even when the learner has access to an optimization oracle for the representation class $\Phi$ [11]. This intractability arises because these algorithms implement optimism via a "global" approach, in which the algorithm explores at each round by choosing the most optimistic value function in a certain *version space* of candidate value functions; optimizing over this version space is challenging,

as it involves satisfying non-convex constraints with a complicated dependence on the learned representation, and because the constraints are coupled globally across layers $h \in [H]$.

To avoid the intractability of global optimism, several works have restricted attention to a simpler *model-based* setting. Here, in addition to assuming that the feature maps $(\phi_h^\star)_{h \in [H]}$ are realizable with respect to $\Phi$, one assumes access to a second feature class $\Upsilon$ capable of modeling the mappings $(\mu_h^\star)_{h \in [H]}$; this facilitates direct estimation of the transition probability kernel $T_h(\cdot \mid x, a)$. For the model-based setting, it is possible to efficiently implement certain "local" forms of optimism [40, 10, 48], as well as certain non-optimistic exploration techniques based on policy covers [1]. Unfortunately, model-based realizability is a restrictive assumption, and falls short of the model-free guarantees we aim for in this work; indeed, in general, one cannot hope to estimate the feature map $\mu_{h+1}^\star$ without sample complexity scaling with the number of states.[4]

When one moves from model-based learning to model-free learning, representation learning becomes substantially more challenging—both for optimistic and non-optimistic approaches. Here, a key challenge is to develop representation learning procedures that are (1) efficient, yet (2) provide meaningful guarantees when the learned features are used downstream for exploration. To our knowledge, the only proposal for a representation learning procedure satisfying both desiderata comes from the work of Modi et al. [35], who introduced a promising "minimax" representation learning objective (described in detail in the sequel; cf. Algorithm 5), which Zhang et al. [49] subsequently showed to have encouraging empirical performance. However, to provide guarantees for this objective, both works place substantial additional restrictions on the low-rank factorization. In particular, Modi et al. [35] make the so-called *latent variable* assumption [1], which asserts that $\phi_h^\star$ and $\mu_h^\star$ are non-negative coordinate-wise, and Zhang et al. [49] further restrict to the Block MDP model [12, 32]. Non-negativity is a substantial restriction, as the best non-negative factorization can have exponentially large dimension relative to the best unrestricted factorization [1], even when reachability is assumed (cf. Appendix H.1). The source of this restriction is the problem of how to quantify how close a learned representation $\hat{\phi}$ is to the ground truth $\phi^\star$, which depends strongly on the downstream exploration strategy. In what follows, we show that with the right exploration strategy, this challenge can be ameliorated, but prior to our work it was unclear whether the minimax objective could lead to meaningful guarantees in the absence of non-negativity.

### 3.2 The SpanRL Algorithm

Our algorithm, SpanRL, is presented in Algorithm 1. The algorithm proceeds by building a policy cover layer-by-layer in an inductive fashion. For each layer $h \geq 2$, SpanRL uses a policy cover $\Psi^{(h)}$ built at a previous iteration within a subroutine, RepLearn (Algorithm 5; deferred to Appendix B) to produce a feature map $\hat{\phi}^{(h)}$ that approximates $\phi_h^\star$. Using this feature map, the algorithm invokes a second subroutine, RobustSpanner (Algorithm 2 in Appendix B) to produce a collection of policies $\pi_1, \ldots, \pi_d$ that act as a *barycentric spanner* for the feature map, ensuring maximal coverage in a certain sense; given these policies, a new policy cover for layer $h + 2$ is formed via $\Psi^{(h+2)} = \{\pi_i \circ_{h+1} \pi_{\text{unif}} : i \in [d]\}$. To invoke the RobustSpanner subroutine, SpanRL makes use of additional subroutines for policy optimization (PSDP; Algorithm 3 in Appendix B) and estimation of certain vector-valued functionals (EstVec; Algorithm 7 in Appendix B). We now describe each component of the algorithm in detail, highlighting how they allow us to overcome the challenges in the prequel.

**Barycentric spanners.** At the heart of SpanRL is the notion of a *barycentric spanner* [6] as an efficient basis for exploration. We begin by defining a barycentric spanner for an abstract set.

**Definition 3.1** (Awerbuch and Kleinberg [6]). *Given a set $\mathcal{W} \subset \mathbb{R}^d$ such that $\text{span}(\mathcal{W}) = \mathbb{R}^d$, we say that a set $\{w_1, \ldots, w_d\} \subseteq \mathcal{W}$ is a $(C, \varepsilon)$-approximate barycentric spanner for $\mathcal{W}$ if for every $w \in \mathcal{W}$, there exist $\beta_1, \ldots, \beta_d \in [-C, C]$ such that $\|w - \sum_{i=1}^{d} \beta_i w_i\| \leq \varepsilon$.[5]*

The utility of barycentric spanners for reward-free exploration is highlighted in the following lemma.

**Lemma 3.1.** *Suppose that Assumption 2.1 holds. If $\Psi \subseteq \Pi_{\mathsf{M}}$ is a collection of policies such that $\{\mathbb{E}^\pi [\phi_h^\star(\boldsymbol{x}_h, \boldsymbol{a}_h)] \mid \pi \in \Psi\} \subseteq \mathbb{R}^d$ is a $(C, \varepsilon)$-approximate barycentric spanner for $\mathcal{W}_h :=\{\mathbb{E}^\pi [\phi_h^\star(\boldsymbol{x}_h, \boldsymbol{a}_h)] \mid \pi \in \Pi_{\mathsf{M}}\}$ with $\varepsilon \leq \frac{\eta}{2}$, then $\Psi$ is an $\alpha$-policy cover for layer $h+1$ with $\alpha = (2dC)^{-1}$.*

---

[4]For example, in the special case of the Block MDP setting [12, 32], model-based realizability entails modeling a certain emission process, which is not required by model-free approaches.

[5]Note that our definition is a slight generalization of [6, Definition 2.1]; the latter is recovered with $\varepsilon = 0$.

Lemma 3.1, proven in Appendix F.6, shows that to compute a policy cover for layer $h+1$, it suffices to find a barycentric spanner for the set $\mathcal{W}_h := \{\mathbb{E}^\pi[\phi_h^\star(\boldsymbol{x}_h, \boldsymbol{a}_h)] \mid \pi \in \Pi_{\mathsf{M}}\} \subseteq \mathbb{R}^d$. Of course, even if $\phi_h^\star$ is known, this observation is only useful if we can compute a spanner without explicitly enumerating over the set $\Pi_{\mathsf{M}}$, since our goal is to develop an *efficient* algorithm. In what follows, we will show:[6]

1. Using, RobustSpanner, a novel adaptation of the classical algorithm of Awerbuch and Kleinberg [6], it holds that for any $\phi \in \Phi$, spanner computation for the set $\{\mathbb{E}^\pi[\phi(\boldsymbol{x}_h, \boldsymbol{a}_h)] \mid \pi \in \Pi_{\mathsf{M}}\}$ can be performed efficiently whenever, for any $\theta \in \mathcal{B}(1)$, one can (approximately) solve linear optimization problems of the form

$$\arg\max_{\pi \in \Pi_{\mathsf{M}}} \mathbb{E}^\pi[\theta^\top \phi(\boldsymbol{x}_h, \boldsymbol{a}_h)]. \tag{5}$$

2. Given access to policy covers $\Psi^{(1:h)}$ for layers 1 to $h$, one can efficiently solve the optimization problem in (5) by appealing to the PSDP algorithm for policy optimization (Algorithm 3).

To handle the fact that $\phi_h^\star$ is unknown, Algorithm 1 computes policies $\pi_{1:d}$ that induce a barycentric spanner for the set $\{\mathbb{E}^\pi[\hat{\phi}^{(h)}(\boldsymbol{x}_h, \boldsymbol{a}_h)] \mid \pi \in \Pi_{\mathsf{M}}\}$, where $\hat{\phi}^{(h)} \in \Phi$ is a learned feature map. In what follows, we first give a detailed explanation of the two above points, before showing how to complete the argument by learning a feature map through representation learning.

---

**Algorithm 1** SpanRL: Volumetric Exploration and Representation Learning via Barycentric Spanner

---

**Require:** Feature class $\Phi$ and parameters $\varepsilon, \mathfrak{c} > 0$ and $\delta \in (0, 1)$.

1: Set $\Psi^{(1)} = \varnothing$, $\Psi^{(2)} = \{\pi_{\mathsf{unif}}\}$.
2: Set $n_{\mathsf{RepLearn}} = \mathfrak{c} \cdot \varepsilon^{-2} A^2 d \log(|\Phi|/\delta)$ and $n_{\mathsf{EstVec}} = \mathfrak{c} \cdot \varepsilon^{-2} \log(1/\delta)$.
3: Set $n_{\mathsf{PSDP}} = \mathfrak{c} \cdot \varepsilon^{-2} A^2 d^3 H^2 \cdot (d + \log(|\Phi|/\delta))$.
4: Define $\mathcal{F} := \{f : x \mapsto \max_{a \in \mathcal{A}} \theta^\top \phi(x, a) \mid \theta \in \mathcal{B}(1), \phi \in \Phi\}$.
5: Define $\mathcal{G} = \{g : (x, a) \mapsto \phi(x, a)^\top w \mid \phi \in \Phi, w \in \mathcal{B}(2\sqrt{d})\}$.
6: **for** $h = 1, \ldots, H - 2$ **do**
        /* Learn feature representation for layer $h$. */
7:     Set $\phi^{(h)} = \mathsf{RepLearn}(h, \mathcal{F}, \Phi, P^{(h)}, n_{\mathsf{RepLearn}})$, with $P^{(h)} = \mathsf{unif}(\Psi^{(h)})$.   // Algorithm 6.
        /* Computing an approximate spanner using learned features. */
8:     For $\theta \in \mathbb{R}^d$ and $(x, a) \in \mathcal{X} \times \mathcal{A}$, define

$$r_t(x, a; \theta) := \begin{cases} \phi^{(h)}(x, a)^\top \theta, & \text{for } t = h, \\ 0, & \text{otherwise.} \end{cases}$$

9:     For each $t \in [h]$, set $\mathcal{G}_t = \mathcal{G}$ and $P^{(t)} = \mathsf{unif}(\Psi^{(t)})$.
10:    For $\theta \in \mathbb{R}^d$, define $\mathsf{LinOpt}(\theta) = \mathsf{PSDP}(h, r_{1:h}(\cdot, \cdot; \theta), \mathcal{G}_{1:h}, P^{(1:h)}, n_{\mathsf{PSDP}})$.   // Algorithm 3.
11:    For $\pi \in \Pi_{\mathsf{M}}$, define $\mathsf{LinEst}(\pi) = \mathsf{EstVec}(h, \phi^{(h)}, \pi, n_{\mathsf{EstVec}})$.   // Algorithm 7.
12:    Set $(\pi_1, \ldots, \pi_d) = \mathsf{RobustSpanner}(\mathsf{LinOpt}(\cdot), \mathsf{LinEst}(\cdot), 2, \varepsilon)$.   // Algorithm 2.
13:    Set $\Psi^{(h+2)} = \{\pi_i \circ_{h+1} \pi_{\mathsf{unif}} : i \in [d]\}$.
14: **Return:** Policy cover $\Psi^{(1:H)}$.

---

**Barycentric spanner computation via approximate linear optimization.** To describe spanner computation in SpanRL, we take a brief detour and consider an abstract approach to barycentric spanner computation, which generalizes our problem. Suppose that we wish to compute a spanner for an implicitly specified set $\mathcal{W} = \{w^z\}_{z \in \mathcal{Z}} \subseteq \mathbb{R}^d$ indexed by an abstract set $\mathcal{Z}$. The set $\mathcal{Z}$ (which will be set to $\Pi_{\mathsf{M}}$ when we return to RL) may be exponentially large and cannot be efficiently enumerated. In addition, given $z \in \mathcal{Z}$, we cannot explicitly compute $w^z$, and have to settle for a noisy approximation.

To allow for efficient spanner computation, we assume access to two oracles for the set $\mathcal{W}$, a *linear optimization* oracle $\mathsf{LinOpt} : \mathcal{B}(1) \to \mathcal{Z}$ and an *index-to-vector* oracle $\mathsf{LinEst} : \mathcal{Z} \to \mathbb{R}^d$. We assume that for some $\varepsilon > 0$:

---

[6]While barycentric spanners have been used in a number of recent works on sample-efficient RL [19, 20], the motivation for their use within our algorithm and analysis are quite different; see Appendix A.

1. For all $\theta \in \mathbb{R}^d$ with $\|\theta\| = 1$, the output $\hat{z}_\theta \coloneqq \mathtt{LinOpt}(\theta)$ satisfies $\theta^\top w^{\hat{z}_\theta} \geq \sup_{z \in \mathcal{Z}} \theta^\top w^z - \varepsilon$.

2. For all $z \in \mathcal{Z}$, the output $\hat{w}_z \coloneqq \mathtt{LinEst}(z)$ satisfies $\|\hat{w}_z - w^z\| \leq \varepsilon$.

The $\mathtt{RobustSpanner}$ algorithm (Algorithm 2) computes a $(C, \varepsilon)$-approximate spanner for $\mathcal{W}$ using $O(d \log(d/\varepsilon))$ total calls to $\mathtt{LinOpt}$ and $\mathtt{LinEst}$. $\mathtt{RobustSpanner}$ is an error-tolerant variant of the classical spanner computation algorithm of Awerbuch and Kleinberg [6], which was originally introduced and analyzed for spanner computation with an *exact* linear optimization oracle. Tolerance to approximation errors in the linear optimization oracle is critical for our application to RL, where additive errors will arise in sampling trajectories, as well as estimating the feature maps $(\phi_h^\star)_{h \in [H]}$. $\mathtt{RobustSpanner}$ achieves error tolerance by perturbing the vectors returned by $\mathtt{LinOpt}(\theta)$ in the direction of $\theta$, which amounts to running the classical algorithm on an $\varepsilon$-fattening of $\mathcal{W}$, and is necessary in order to ensure that the approximation error of $\mathtt{LinOpt}$ does not swamp the signal in directions $\theta$ in which $\mathcal{W}$ is too "skinny." This technique may be of independent interest; see Appendix C for additional details and formal guarantees.

**Representation learning.** Ideally, we would like to use $\mathtt{RobustSpanner}$ to construct a barycentric spanner for the set $\{\mathbb{E}^\pi[\phi_h^\star(\boldsymbol{x}_h, \boldsymbol{a}_h)] \mid \pi \in \Pi_\mathsf{M}\}$ with $\mathcal{Z} = \Pi_\mathsf{M}$. Because we do not have access to $\phi_h^\star$, we instead apply $\mathtt{RobustSpanner}$ with $\mathcal{W} \coloneqq \{\mathbb{E}^\pi[\hat{\phi}^{(h)}(\boldsymbol{x}_h, \boldsymbol{a}_h)] \mid \pi \in \Pi_\mathsf{M}\}$, where $\hat{\phi}^{(h)}$ is a learned representation. We now describe how the feature map $\hat{\phi}^{(h)}$ is learned, then show how to use these learned features to efficiently implement the oracles $\mathtt{LinOpt}(\cdot)$ and $\mathtt{LinEst}(\cdot)$.

To learn a representation for layer $h$, we use the $\mathtt{RepLearn}$ algorithm (Algorithm 5), which was originally introduced in Modi et al. [35], Zhang et al. [49]. The algorithm gathers a collection of triples $(\boldsymbol{x}_h, \boldsymbol{a}_h, \boldsymbol{x}_{h+1})$ by rolling in to $\boldsymbol{x}_h$ with a policy sampled uniformly from the policy cover $\Psi^{(h)}$ and selecting $\boldsymbol{a}_h$ uniformly at random. Using this dataset, the algorithm solves a sequence of adversarial training sub-problems (Line 9 of Algorithm 5) which involve the feature class $\Phi$ and an auxiliary discriminator class $\mathcal{F} : \mathcal{X} \to \mathbb{R}$. As we discuss in detail in the sequel, these sub-problems, described in (7), are amenable to standard gradient-based training methods. The sub-problems are designed to approximate the following "idealized" max-min-max representation learning objective:

$$\hat{\phi}^{(h)} \in \arg\min_{\phi \in \Phi} \sup_{f \in \mathcal{F}} \inf_w \mathbb{E}^{\mathsf{unif}(\Psi^{(h)}) \circ_h \pi_{\mathsf{unif}}}\left[\left(\phi(\boldsymbol{x}_h, \boldsymbol{a}_h)^\top w - \mathbb{E}[f(\boldsymbol{x}_{h+1}) \mid \boldsymbol{x}_h, \boldsymbol{a}_h]\right)^2\right]. \quad (6)$$

The intuition for this objective comes from the fact that in a Low-Rank MDP, for any function $f : \mathcal{X} \to \mathbb{R}$, the quantity $\mathbb{E}[f(\boldsymbol{x}_{h+1}) \mid \boldsymbol{x}_h = x, \boldsymbol{a}_h = a]$ is linear in $\phi_h^\star(x, a)$. Thus, if $\mathcal{F}$ is sufficiently expressive, we may hope that $\hat{\phi}^{(h)}$ and $\phi^\star$ are close. We adopt the simple discriminator class $\mathcal{F} = \{x \mapsto \max_{a \in \mathcal{A}} \theta^\top \phi(x, a) \mid \theta \in \mathcal{B}(1), \phi \in \Phi\}$. We show that solving (6) with this choice for $\mathcal{F}$, which is simpler than that in Modi et al. [35], Zhang et al. [49], yields an approximation guarantee for $\hat{\phi}^{(h)}$ that is suitable for downstream use in spanner computation for general Low-Rank MDPs.

**Remark 3.1** (Improved analysis of $\mathtt{RepLearn}$). *To facilitate an analysis of $\mathtt{SpanRL}$ that does not require reachability assumptions, we use slightly different parameter values for $\mathtt{RepLearn}$ than in Modi et al. [35], Zhang et al. [49], and provide a tighter sample complexity bound (Theorem E.1) which may be of independent interest.*

*In more detail, prior work shows that the $\mathtt{RepLearn}$ algorithm solves a variant of (6) with $w \in \mathcal{B}(d^{1/2} \cdot \mathrm{poly}(\varepsilon^{-1}))$, where $\varepsilon > 0$ is the desired bound on mean-squared error. Due to the polynomial dependence on $\varepsilon^{-1}$, such a guarantee would lead to vacuous guarantees when invoked within our analysis of $\mathtt{SpanRL}$. Our improved analysis of $\mathtt{RepLearn}$, which is based on a determinantal potential argument, shows that $w \in \mathcal{B}(\mathrm{poly}(d))$ suffices. A secondary benefit of our improved bound is a faster rate with respect to the number of trajectories.*

**Putting everything together.** Having learned $\hat{\phi}^{(h)}$ using $\mathtt{RepLearn}$, in $\mathtt{SpanRL}$ we apply $\mathtt{RobustSpanner}$ with $\mathcal{W} \coloneqq \{\mathbb{E}^\pi[\hat{\phi}^{(h)}(\boldsymbol{x}_h, \boldsymbol{a}_h)] \mid \pi \in \Pi_\mathsf{M}\}$, $\mathcal{Z} = \Pi_\mathsf{M}$, and $C = 2$; that is, we plug-in the learned representation $\hat{\phi}^{(h)}$ for the true representation $\phi_h^\star$.[7] With this choice, implementing $\mathtt{LinOpt}$ entails (approximately) solving $\arg\max_{\pi \in \Pi_\mathsf{M}} \mathbb{E}^\pi[\theta^\top \hat{\phi}^{(h)}(\boldsymbol{x}_h, \boldsymbol{a}_h)]$ for a given $\theta \in \mathcal{B}(1)$, and implementing the $\mathtt{LinEst}$ oracle entails estimating $\mathbb{E}^\pi[\hat{\phi}^{(h)}(\boldsymbol{x}_h, \boldsymbol{a}_h)]$ for a given $\pi \in \Pi_\mathsf{M}$. We instantiate

---

[7]Though the policies produced by the algorithm may not necessarily induce a spanner for $\mathcal{W}_h = \{\mathbb{E}^\pi[\phi_h^\star(\boldsymbol{x}_h, \boldsymbol{a}_h)] \mid \pi \in \Pi_\mathsf{M}\}$ (this would require "point-wise" representation learning guarantees, which we do not have), our analysis shows that they still suffice to build a policy cover for layer $h + 2$.

LinEst($\pi$) as the Monte Carlo algorithm EstVec (Algorithm 7), which simply samples trajectories according to $\pi$ and returns the sample average of $\hat{\phi}^{(h)}(x_h, a_h)$. To implement LinOpt($\theta$), we appeal to PSDP (Algorithm 3). PSDP, given an arbitrary reward function $r_{1:h} : \mathcal{X} \times \mathcal{A} \to \mathbb{R}$ and a function class $\mathcal{G} \subseteq \{g : \mathcal{X} \times \mathcal{A} \to \mathbb{R}\}$ capable of realizing all possible value functions induced by these rewards, can use the policy covers $\Psi^{(1:h)}$ to efficiently compute a policy $\hat{\pi} = \text{PSDP}(h, r_{1:h}, \mathcal{G}, \Psi^{(1:h)}, n)$ that approximately solves $\arg\max_{\pi \in \Pi_M} \mathbb{E}^{\pi}[\sum_{t=1}^{h} r_t(x_t, a_t)]$, and does so using polynomially many episodes; see Appendix D for details and formal guarantees.[8] Thus, implementing LinOpt($\theta$) is as simple as invoking PSDP with the rewards

$$r_t(x, a; \theta) \coloneqq \begin{cases} \hat{\phi}^{(h)}(x, a)^{\top}\theta, & \text{for } t = h, \\ 0, & \text{otherwise.} \end{cases}$$

With this, we have all the ingredients needed for spanner computation, and the algorithm is complete.

### 3.3 Main Guarantee for SpanRL

The following result is the main sample complexity guarantee for SpanRL (Algorithm 1).

**Theorem 3.2** (Main theorem for SpanRL). *Let $\delta \in (0, 1)$ be given, and suppose that realizability holds (Assumption 2.2) and that reachability (Assumption 2.1) is satisfied with parameter $\eta > 0$. If $\varepsilon = \frac{\eta}{36d^{5/2}}$ and $\mathfrak{c} = \text{polylog}(A, H, d, \log(|\Phi|/\delta))$ is sufficiently large, then the policies $\Psi^{(1:H)}$ produced by SpanRL$(\Phi, \varepsilon, \mathfrak{c}, \delta)$ are a $(\frac{1}{4Ad}, 0)$-policy cover with probability at least $1 - \delta$. The total number of episodes used by SpanRL is at most:*

$$\widetilde{O}\left(A^4 d^9 H^4 (d + \log(|\Phi|/\delta)) \cdot 1/\eta^2\right).$$

Theorem 3.2 is the first provable, model-free sample complexity guarantee for general Low-Rank MDPs that is attained by an efficient algorithm. Prior to our work, all efficient model-free algorithms required non-negative features (latent variable structure) or stronger assumptions [35, 49], even in the presence of similar reachability assumptions; see Table 1.

**Remark 3.2** (On the reachability assumption). *While the reachability assumption is shared by the best prior efficient algorithms [35], which require non-negativity in addition to this assumption, it is natural to ask to what extent reachability restricts the generality of the Low-Rank MDP model. In Appendix H, we show that even when reachability holds, the embedding dimension in our model can be exponentially smaller than the best embedding dimension for the best non-negative (latent variable) embedding [35]. Hence, our results are meaningfully more general than prior work.*

While our guarantee is polynomial in all relevant problem parameters, improving the dependence further (e.g., to match that of the best known inefficient algorithms) is an interesting direction for future research, as is removing the reachability assumption.

**Application to reward-based RL.** By using the policy cover produced by SpanRL within PSDP (Algorithm 3), we can optimize any downstream reward function to error $\varepsilon$ using $\text{poly}(d, A, H, \log|\Phi|) \cdot 1/\varepsilon^2$ episodes. See Appendix G for details.

**Efficiency and practicality.** We observe that SpanRL is simple and practical. Defining $\mathcal{L}_{\mathcal{D}}(\phi, w, f) \coloneqq \sum_{(x,a,x') \in \mathcal{D}}(\phi(x, a)^{\top}w - f(x'))^2 + \lambda\|w\|^2$, where $\mathcal{D}$ is a dataset consisting of $(x_h, a_h, r_h, x_{h+1})$ tuples, the algorithm is provably efficient whenever the adversarial objective

$$f^{(t)} \in \arg\max_{f \in \mathcal{F}} \max_{\tilde{\phi} \in \Phi} \left\{ \min_{w} \mathcal{L}_{\mathcal{D}}(\phi^{(t)}, w, f) - \min_{\tilde{w}} \mathcal{L}_{\mathcal{D}}(\tilde{\phi}, \tilde{w}, f) \right\}, \tag{7}$$

in Line 9 of RepLearn (Algorithm 5), can be implemented efficiently (note that by the definition of $\mathcal{L}_{\mathcal{D}}$, the "inner" minima over $w$ and $\tilde{w}$ in (7) can be solved in closed form). This objective was also assumed to be efficiently solvable in Modi et al. [35], Zhang et al. [49] and was empirically shown to be practical in [49]; note that the objective is amenable to standard gradient-based optimization techniques, and that $\mathcal{F}$ can be over-parameterized. While a detailed experimental evaluation is outside of the scope of this paper, we are optimistic about the empirical performance of the algorithm in light of the encouraging results based on the same objective in Zhang et al. [49]

---

[8]This is the main place where the analysis uses the inductive hypothesis that $\Psi^{(1:h)}$ are policy covers.

Outside of representation learning, the only overhead in SpanRL is the RobustSpanner subroutine, which has polynomial runtime. Indeed, RobustSpanner requires only polynomially many calls to the linear optimization oracle, instantiated as PSDP, which is efficient whenever standard least-squares regression problems based on the class $\Phi$ can be solved efficiently, analogous to [33, 31].

**Analysis and proof techniques.** The proof of Theorem 3.2, which is given in Appendix F, is appealing in its simplicity and modularity. The crux of the proof is to show that the representation learning guarantee in (6) is strong enough to ensure that the downstream spanner computation in RobustSpanner succeeds. It is straightforward to show that spanner computation would succeed if we had access to an estimated representation that $\hat{\phi}^{(h)}$ that approximates $\phi_h^\star$ point-wise (i.e., uniformly for all $(x, a)$ pairs), but the key challenge is that the guarantee in (6) only holds *on average* under the roll-in distribution $\mathrm{unif}(\Psi^{(h)})$. Prior works that make use of the same representation learning objective (BRIEE [49] and MOFFLE [35]) do not make use of spanners; instead, they appeal to exploration strategies based on elliptic bonuses, addressing the issue of approximation errors through additional assumptions (non-negativity of the factorization for MOFFLE, and Block MDP structure for BRIEE). Perhaps the most important observation in our proof is that barycentric spanners are robust to the average-case approximation error guarantee in (6) as-is, without additional structural assumptions. Intuitively, this benefit seems to arise from the fact that the spanner property only concerns the *first moment* of the feature map $\phi^\star$, while algorithms based on elliptic bonuses require approximation guarantees for the *second moment*; understanding this issue more deeply is an interesting question for future work. Another useful feature of our proof is to show that the notion of reachability in Assumption 2.1, which generalizes and extends all previous reachability conditions in the Low-Rank MDP and Block MDP literature [46, 5, 13, 33, 1, 35, 31], is sufficient to build a policy cover. We anticipate that this observation will find broader use.

## Acknowledgments and Disclosure of Funding

We thank Noah Golowich, Dhruv Rohatgi, and Ayush Sekhari for several helpful discussions. ZM and AR acknowledge support from the ONR through awards N00014-20-1-2336 and N00014-20-1-2394, and ARO through award W911NF-21-1-0328. AB acknowledges support from the National Science Foundation Graduate Research Fellowship under Grant No. 1122374.

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

## Contents of Appendix

## A Additional Related Work

In this section, we discuss relevant related work not already covered.

**Block MDPs.** A particularly well-studied special case low-rank MDPs with the latent variable assumed in Modi et al. [35] (defined in Definition H.1) is the *Block MDP (BMDP) model* Du et al. [13], Misra et al. [32], Zhang et al. [49], Mhammedi et al. [31]. For this setting, Du et al. [13], Misra et al. [32] provide algorithms that conduct exploration in a provably oracle-efficient manner under a reachability assumption. This reachability assumption was removed by subsequent work of Zhang et al. [49] (with a suboptimal rate) and Mhammedi et al. [31] (with optimal error dependence), but the analysis in these works is tailored to the BMDP model.

**Barycentric spanners.** Huang et al. [20] consider a variant of the Low-Rank MDP framework in which we are given a class $\Upsilon$ that realizes the next-state feature map $\mu^\star$, but do not have access to a class $\Phi$ for the feature map $\phi^\star$, which is unknown. Their algorithm, like `SpanRL`, is based on barycentric spanners, though the algorithm design considerations and analysis are significantly different. Notably, their algorithm is not computationally efficient, and their analysis takes advantage of the fact that realizability of $\mu^\star$ facilitates estimation of the occupancies $\{d^\pi(\cdot)\}_{\pi \in \Pi_\mathsf{M}}$ in $\ell_1$-error. Barycentric spanners were also in the work of Golowich et al. [19] for reinforcement learning in Partially Observable MDPs (POMDPs). Their analysis is substantially different from ours, and their algorithm appeals to the barycentric spanner computation approach in Awerbuch and Kleinberg [6] in an off-the-shelf fashion.

# B  Omitted Algorithms

---

**Algorithm 2** RobustSpanner: Barycentric Spanner via Approximate Linear Optimization

---

**Require:**

- Approximate linear optimization subroutine $\mathtt{LinOpt}: \mathbb{R}^d \to \mathcal{Z}$.    `/* See Section 3.2 */`
- Approximate index-to-vector subroutine $\mathtt{LinEst}: \mathcal{Z} \to \mathbb{R}^d$.
- Parameters $C, \varepsilon > 0$.

1: Set $W = (w_1, \ldots, w_d) = (e_1, \ldots, e_d)$.
2: **for** $i = 1, \ldots, d$ **do**
3:     Set $\theta_i = (\det(e_j, W_{-i}))_{j \in [d]} \in \mathbb{R}^d$.  `// `$W_{-i}$` is defined to be `$W$` without the `$i$`th column`
4:     Set $z_i^+ = \mathtt{LinOpt}(\theta_i/\|\theta_i\|)$ and $w_i^+ = \mathtt{LinEst}(z_i^+)$.
5:     Set $z_i^- = \mathtt{LinOpt}(-\theta_i/\|\theta_i\|)$ and $w_i^- = \mathtt{LinEst}(z_i^-)$.
6:     **if** $\theta_i^\top w_i^+ \geq -\theta_i^\top w_i^-$ **then**
7:         Set $\widetilde{w}_i = w_i^+$, $z_i = z_i^+$, and $w_i = \widetilde{w}_i + \varepsilon \theta_i/\|\theta_i\|$.
8:     **else**
9:         Set $\widetilde{w}_i = w_i^-$, $z_i = z_i^-$, and $w_i = \widetilde{w}_i - \varepsilon \theta_i/\|\theta_i\|$.
10: **for** $n = 1, 2, \ldots$ **do**
11:     Set $i = 1$.
12:     **while** $i \leq d$ **do**
13:         Set $\theta_i = (\det(e_j, W_{-i}))_{j \in [d]} \in \mathbb{R}^d$.
14:         Set $z_i^+ = \mathtt{LinOpt}(\theta_i/\|\theta_i\|)$ and $w_i^+ = \mathtt{LinEst}(z_i^+)$.
15:         Set $z_i^- = \mathtt{LinOpt}(-\theta_i/\|\theta_i\|)$ and $w_i^- = \mathtt{LinEst}(z_i^-)$.
16:         **if** $\theta_i^\top w_i^+ + \varepsilon \cdot \|\theta_i\| \geq C \cdot |\det(w_i, W_{-i})|$ **then**
17:             Set $\widetilde{w}_i = w_i^+$, $z_i = z_i^+$, and $w_i = \widetilde{w}_i + \varepsilon \cdot \theta_i/\|\theta_i\|$.
18:             **break**
19:         **else if** $-\theta_i^\top w_i^- + \varepsilon \cdot \|\theta_i\| \geq C \cdot |\det(w_i, W_{-i})|$ **then**
20:             Set $\widetilde{w}_i = w_i^-$, $z_i = z_i^-$, and $w_i = \widetilde{w}_i - \varepsilon \cdot \theta_i/\|\theta_i\|$.
21:             **break**
22:         Set $i = i + 1$.
23:     **if** $i = d + 1$ **then**
24:         **break**
25: **Return:** $(z_1, \ldots, z_d)$.

---

**Algorithm 3** $\mathtt{PSDP}(h, r_{1:h}, \mathcal{G}, \Psi^{(1:h)}, n)$: Policy Search by Dynamic Programming (variant of [8])

**Require:**

- Target layer $h \in [H]$.
- Reward functions $r_{1:h}$.
- Function class $\mathcal{G}$.
- Policy covers $\Psi^{(1)}, \ldots, \Psi^{(h)}$.
- Number of samples $n \in \mathbb{N}$.

1: **for** $t = h, \ldots, 1$ **do**

2: $\quad \mathcal{D}^{(t)} \leftarrow \varnothing$.

3: $\quad$ **for** $n$ times **do**

4: $\quad\quad$ Sample $(\boldsymbol{x}_t, \boldsymbol{a}_t, \sum_{\ell=t}^{h} r_\ell(\boldsymbol{x}_\ell, \boldsymbol{a}_\ell)) \sim \mathtt{unif}(\Psi^{(t)}) \circ_t \pi_{\mathtt{unif}} \circ_{t+1} \hat{\pi}^{(t+1)}$.

5: $\quad\quad$ Update dataset: $\mathcal{D}^{(t)} \leftarrow \mathcal{D}^{(t)} \cup \left\{ \left( \boldsymbol{x}_t, \boldsymbol{a}_t, \sum_{\ell=t}^{h} r_\ell(\boldsymbol{x}_\ell, \boldsymbol{a}_\ell) \right) \right\}$.

6: $\quad$ Solve regression:
$$g^{(t)} \leftarrow \operatorname*{arg\,min}_{g \in \mathcal{G}} \sum_{(x,a,R) \in \mathcal{D}^{(t)}} (g(x,a) - R)^2.$$

7: $\quad$ Define $\hat{\pi}^{(t)} \in \Pi_{\mathsf{M}}^{t:h}$ via

$$\hat{\pi}^{(t)}(x) = \begin{cases} \arg\max_{a \in \mathcal{A}} g^{(t)}(x,a), & x \in \mathcal{X}_t, \\ \hat{\pi}^{(t+1)}(x), & x \in \mathcal{X}_\ell, \ \ell \in [t+1 \mathinner{..} h]. \end{cases} \quad (8)$$

8: **Return:** Near-optimal policy $\hat{\pi}^{(1)} \in \Pi_{\mathsf{M}}$.

---

**Algorithm 4** $\mathtt{EstVec}(h, F, \pi, n)$: Estimate $\mathbb{E}^\pi[F(\boldsymbol{x}_h, \boldsymbol{a}_h)]$ for policy $\pi$ and function $F : \mathcal{X} \times \mathcal{A} \to \mathbb{R}^d$.

**Require:**

- Target layer $h \in [H]$.
- Vector-valued function $F : \mathcal{X} \times \mathcal{A} \to \mathbb{R}^d$.
- Policy $\pi \in \Pi_{\mathsf{M}}$.
- Number of samples $n \in \mathbb{N}$.

1: $\mathcal{D} \leftarrow \varnothing$.

2: **for** $n$ times **do**

3: $\quad$ Sample $(\boldsymbol{x}_h, \boldsymbol{a}_h) \sim \pi$.

4: $\quad$ Update dataset: $\mathcal{D} \leftarrow \mathcal{D} \cup \{(\boldsymbol{x}_h, \boldsymbol{a}_h)\}$.

5: **Return:** $\bar{F} = \frac{1}{n} \sum_{(x,a) \in \mathcal{D}} F(x,a)$.

**Algorithm 5** RepLearn($h, \mathcal{F}, \Phi, \Psi, n$): Representation Learning for Low-Rank MDPs [35, 49].

**Require:**
- Target layer $h \in [H]$.
- Discriminator class $\mathcal{F}$.
- Feature class $\Phi$.
- A set of policies $\Psi$.
- Number of samples $n \in \mathbb{N}$.

1: Set $\tilde{\varepsilon} = 100 \frac{d \log n + \log(|\Phi|/\delta)}{n}$, $\varepsilon_1 = 16\sqrt{2} d^{3/2} \tilde{\varepsilon}^{1/2}$.

2: Initialize $\lambda = \sqrt{n}$, $T = \sqrt{\frac{d}{2\tilde{\varepsilon}}}$, and $\xi = \frac{3}{2}\varepsilon_1 + \tilde{\varepsilon} + \frac{2\lambda d}{n}$, $\phi^{(0)} \in \Phi$.

3: $\mathcal{D} \leftarrow \varnothing$,

4: **for** $n$ times **do**

5:      Sample $(\boldsymbol{x}_h, \boldsymbol{a}_h, \boldsymbol{x}_{h+1}) \sim \mathtt{unif}(\Psi) \circ_h \pi_{\mathtt{unif}}$.

6:      Update dataset: $\mathcal{D} \leftarrow \mathcal{D} \cup \{(\boldsymbol{x}_h, \boldsymbol{a}_h, \boldsymbol{x}_{h+1})\}$.

7: Define $\mathcal{L}_{\mathcal{D}}(\phi, w, f) = \sum_{(x,a,x') \in \mathcal{D}} (\phi(x,a)^{\top} w - f(x'))^2 + \lambda \|w\|^2$.

8: **for** $t = 0, \ldots, T - 1$ **do**

     /* Discriminator selection */

9:      Solve

$$f^{(t)} \in \arg\max_{f \in \mathcal{F}} \widehat{\Delta}(f), \quad \text{where} \quad \widehat{\Delta}(f) := \max_{\tilde{\phi} \in \Phi} \left\{ \min_w \mathcal{L}_{\mathcal{D}}(\phi^{(t)}, w, f) - \min_{\tilde{w}} \mathcal{L}_{\mathcal{D}}(\tilde{\phi}, \tilde{w}, f) \right\}.$$

10:      **if** $\widehat{\Delta}(f^{(t)}) \leq \xi$ **then**

11:          Return $\hat{\phi} = \phi^{(t)}$.

     /* Feature selection via least-squares minimization */

12:      Solve

$$\phi^{(t+1)} \in \arg\min_{\phi \in \Phi} \min_{w_{0:t}} \sum_{i=0}^{t} \mathcal{L}_{\mathcal{D}}(\phi, w_i, f^{(i)}).$$

# C Generic Guarantee for RobustSpanner

In this section, we give a generic guarantee for the RobustSpanner algorithm when invoked with oracles LinOpt and LinEst satisfying the following assumption.

**Assumption C.1** (LinOpt and LinEst as approximate Linear Optimization Oracles). *For some abstract set $\mathcal{Z}$ and a collection of vectors $\{w^z \in \mathbb{R}^d \mid z \in \mathcal{Z}\}$ indexed by elements in $\mathcal{Z}$, there exists $\varepsilon' > 0$ such that for any $\theta \in \mathbb{R}^d \setminus \{0\}$ and $z \in \mathcal{Z}$, the outputs $\hat{z}_\theta \coloneqq \mathtt{LinOpt}(\theta/\|\theta\|)$ and $\hat{w}_z \coloneqq \mathtt{LinEst}(z)$ satisfy*

$$\sup_{z \in \mathcal{Z}} \theta^\top w^z \le \theta^\top w^{\hat{z}_\theta} + \varepsilon' \cdot \|\theta\|, \quad \text{and} \quad \|\hat{w}_z - w^z\| \le \varepsilon'.$$

Letting $\mathcal{W} \coloneqq \{w^z \mid z \in \mathcal{Z}\}$ and assuming that $\mathcal{W} \subseteq \mathcal{B}(1)$, the next theorem bounds the number of iterations of RobustSpanner($\mathtt{LinOpt}(\cdot), \mathtt{LinEst}(\cdot), \cdot, \cdot$) under Assumption C.1, and shows that the output is an approximate barycentric spanner for $\mathcal{W}$ (Definition 3.1). Our result extends those of Awerbuch and Kleinberg [6], in that it only requires an *approximate* linear optimization oracle, which is potentially of independent interest.

**Proposition C.1.** *Fix $C > 1$ and $\varepsilon \in (0, 1)$ and suppose that $\{w^z \mid z \in \mathcal{Z}\} \subseteq \mathcal{B}(1)$. If RobustSpanner (Algorithm 2) is run with parameters $C, \varepsilon > 0$ and oracles LinOpt, LinEst satisfying Assumption C.1 with $\varepsilon' = \varepsilon/2$, then it terminates after $d + \lceil \frac{d}{2} \log_C \frac{100d}{\varepsilon^2} \rceil$ iterations, and requires at most twice that many calls to each of LinOpt and LinEst. Furthermore, the output $z_{1:d}$ has the property that for all $z \in \mathcal{Z}$, there exist $\beta_1, \ldots, \beta_d \in [-C, C]$, such that*

$$\left\| w^z - \sum_{i=1}^d \beta_i w^{z_i} \right\| \le \frac{3Cd \cdot \varepsilon}{2}.$$

**Proof of Proposition C.1.** The proof will follows similar steps to those in Awerbuch and Kleinberg [6, Lemma 2.6], with modifications to account for the fact that linear optimization over the set $\mathcal{W} \coloneqq \{w^z \mid z \in \mathcal{Z}\}$ is only performed approximately.

**Part I: Bounding the number of iterations.** In Algorithm 2, there are two loops, both of which require two calls to LinOpt and LinEst per iteration. As the first loop has exactly $d$ iterations, it suffices to bound the number of iterations in the second loop.

Let $M^{(i)} \coloneqq (w_1, \ldots, w_i, e_{i+1}, \ldots, e_d)$ be the matrix whose columns are the vectors at end of the $i$th iteration of the first loop (Line 2) of Algorithm 2; note that columns $i + 1$ through $d$ are unchanged at this point in the algorithm. For $i \in [d]$, we define $\ell_i(w) \coloneqq \det(w, M_{-i}^{(i)})$ and $\theta_i \coloneqq \left( \det(e_j, M_{-i}^{(i)}) \right)_{j \in [d]} \in \mathbb{R}^d$, where we recall that for any matrix $A$, the matrix $A_{-i}$ is defined as the result of removing the $i$th column from $A$. Note that $\ell_i$ is linear in $w$, and in particular

$$\ell_i(w) \coloneqq w^\top \theta_i.$$

Let $W^{(0)} \coloneqq M^{(d)} = (w_1, \ldots, w_d)$, and let $W^{(j)}$ denote the resulting matrix after $j$ iterations of the second loop (Line 10) of Algorithm 2. We will show that for any $J \ge 1$,

$$\det(W^{(J)}) \le \det(W^{(0)}) \cdot \left( \frac{100d}{\varepsilon^2} \right)^{\frac{d}{2}}. \tag{9}$$

By construction of the loop, we have $\det(W^{(j)}) \ge C \cdot \det(W^{(j-1)})$ for each $j \in [J]$, and thus $\det(W^{(J)}) \ge \det(W^{(0)}) \cdot C^J$. Combining these two facts will establish the bound on the iteration complexity. We now prove (9).

Let $u_i = e_i^\top \left( M^{(i)} \right)^{-1}$ (note that $u_i$ is a *row vector*) and let $U$ denote the matrix whose $i$th row is $u_i$. We observe that for all $w \in \mathbb{R}^d$,

$$u_i w = \frac{\ell_i(w)}{\ell_i(w_i)},$$

where we note that $\ell_i(w_i) \ne 0$ by construction; indeed, the columns of $M^{(i)}$ are a basis for $\mathbb{R}^d$ because $\det(M^{(i)}) \ne 0$, and the equality holds on the columns, so the two linear functions must be equal. Now, since Assumption C.1 holds with $\varepsilon' = \varepsilon/2$, we have

$$\theta_i^\top w_i^+ \ge \sup_{z \in \mathcal{Z}} \theta_i^\top w^z - \frac{\varepsilon}{2}\|\theta_i\|, \quad \text{and} \quad \theta_i^\top w_i^- \le \inf_{z \in \mathcal{Z}} \theta_i^\top w^z + \frac{\varepsilon}{2}\|\theta_i\|, \tag{10}$$

where $w_i^\pm = \texttt{LinOpt}(z_i^\pm)$. We will now show that

$$\ell_i(w_i) \ge \frac{\varepsilon}{2} \cdot \|\theta_i\|. \tag{11}$$

There are two cases. First, suppose that $\theta_i^\top w_i^+ \ge -\theta_i^\top w_i^-$, corresponding to the conditional in Line 6 of Algorithm 2 being satisfied. Combining this with (10), we have

$$\begin{aligned}
\theta_i^\top w_i^+ &\ge \left(\sup_{z \in \mathcal{Z}} \theta_i^\top w^z - \frac{\varepsilon}{2}\|\theta_i\|\right) \vee (-\theta_i^\top w_i^-), \\
&\ge \left(\sup_{z \in \mathcal{Z}} \theta_i^\top w^z - \frac{\varepsilon}{2}\|\theta_i\|\right) \vee \left(\sup_{z \in \mathcal{Z}} -\theta_i^\top w^z - \frac{\varepsilon}{2}\|\theta_i\|\right), \quad \text{(by (10))} \\
&= \left(\sup_{z \in \mathcal{Z}} \theta_i^\top w^z\right) \vee \left(\sup_{z \in \mathcal{Z}} -\theta_i^\top w^z\right) - \frac{\varepsilon}{2}\|\theta_i\|, \\
&\ge -\frac{\varepsilon}{2}\|\theta_i\|. \tag{12}
\end{aligned}$$

Because the conditional is satisfied, $w_i = w_i^+ + \varepsilon \cdot \frac{\theta_i}{\|\theta_i\|}$, and so by plugging this into (12), we have

$$\ell_i(w_i) = \theta_i^\top w_i \ge \frac{\varepsilon}{2} \cdot \|\theta_i\|.$$

The case that $\theta_i^\top w_i^+ \le -\theta_i^\top w_i^-$ is essentially identical, establishing (11). Now, recall that $\mathcal{W} \coloneqq \{w^z \mid z \in \mathcal{Z}\}$ and let $\mathcal{W} \oplus \mathcal{B}\left(\frac{3\varepsilon}{2}\right) \coloneqq \{w + b \mid w \in \mathcal{W} \text{ and } b \in \mathcal{B}\left(\frac{3\varepsilon}{2}\right)\}$ denote the Minkowski sum with $\mathcal{B}\left(\frac{3\varepsilon}{2}\right)$. By Cauchy-Schwarz, it holds that for all $w' \coloneqq w + b \in \mathcal{W} \oplus \mathcal{B}\left(\frac{3\varepsilon}{2}\right)$,

$$\ell_i(w') = \theta_i^\top w' = \theta_i^\top w + \theta_i^\top b \le \left(1 + \frac{3\varepsilon}{2}\right) \cdot \|\theta_i\|,$$

where we used that $\mathcal{W} \subseteq \mathcal{B}(1)$ (by assumption). Thus, for any $w' \in \mathcal{W} \oplus \mathcal{B}\left(\frac{3\varepsilon}{2}\right)$, we have

$$|u_i w'| = \frac{\ell_i(w')}{\ell_i(w_i)} \le 1 + \frac{3\varepsilon}{2}.$$

We now observe that by construction and the fact that Assumption C.1 holds with $\varepsilon' = \varepsilon/2$, the $k$th column $w_k'$ of $W^{(J)}$ belongs to $\mathcal{W} \oplus \mathcal{B}\left(\frac{3\varepsilon}{2}\right)$, for any $k \in [d]$. Thus, the $(i,k)$ entry $u_i w_k'$ of $UW^{(J)}$ satisfies $u_i w_k' \in \left[-1 - \frac{3\varepsilon}{2}, 1 + \frac{3\varepsilon}{2}\right]$, and so the columns of $UW^{(J)}$ have Euclidean norm at most $\frac{10\sqrt{d}}{\varepsilon}$. Since the magnitude of the determinant of a matrix is upper bounded by the product of the Euclidean norms of its columns, it holds that $|\det(UW^{(J)})| \le \left(\frac{100d}{\varepsilon^2}\right)^{\frac{d}{2}}$.

On the other hand, again by construction, we see that the columns $w_1, \dots, w_d$ of $W^{(0)}$ satisfy $u_i w_j = 0$, for $j < i$, and $u_i w_i = 1$. Thus, $UW^{(0)}$ is an upper-triangular matrix with 1s on the diagonal, and hence has determinant 1. Because determinants are multiplicative, this implies that $\det(U) \ne 0$. We now compute:

$$|\det(W^{(J)})| = \frac{|\det(UW^{(J)})|}{|\det(U)|} = \frac{|\det(UW^{(J)})|}{|\det(UW^{(0)})|} \le \left(\frac{100d}{\varepsilon^2}\right)^{\frac{d}{2}}.$$

Thus, the upper bound on $|\det(W^{(J)})|$ holds and the claim is proven. Therefore, we have

$$C^J \le \left(\frac{100d}{\varepsilon^2}\right)^{\frac{d}{2}},$$

and so $J \le \left\lceil \frac{d}{2} \log_C\left(\frac{100d}{\varepsilon^2}\right)\right\rceil$.

**Part II: Spanner property for the output.**  Having shown that the algorithm terminates, we now show that the result is an approximate barycentric spanner for $\mathcal{W}$. Let $W \coloneqq (w_1, \ldots, w_d)$ be the matrix at termination of the algorithm. By definition, if the second loop (Line 10) has terminated, then for all $i \in [d]$,

$$\max(\theta_i^\top w_i^+, -\theta_i^\top w_i^-) + \varepsilon \cdot \|\theta_i\| \leq C \cdot |\det(w_i, W_{-i})|,$$

where $\theta_i = (\det(e_j, W_{-i}))_{j \in [d]} \in \mathbb{R}^d$. On the other hand, by Assumption C.1, (10) holds, and so

$$\forall z \in \mathcal{Z}, \forall i \in [d], \quad |\det(w^z, W_{-i})| = |\theta_i^\top w^z| \leq \max(\theta_i^\top w_i^+, -\theta_i^\top w_i^-) + \varepsilon \cdot \|\theta_i\|, \\ \leq C \cdot |\det(w_i, W_{-i})|. \tag{13}$$

Now, fix $z \in \mathcal{Z}$. Since $\det(W) \neq 0$, there exist $\beta_{1:d} \in \mathbb{R}$ such that $w^z = \sum_{i=1}^d \beta_i w_i$. By plugging this into (13) and using the linearity of the determinant, we have

$$\forall i \in [d], \quad C \cdot |\det(w_i, W_{-i})| \geq |\det(w^z, W_{-i})| = \left| \sum_{j=1}^d \beta_i \det(w_j, W_{-i}) \right| = |\beta_i| \cdot |\det(w_i, W_{-i})|.$$

Therefore, $|\beta_i| \leq C$, for all $i \in [d]$. Now, by definition of $w_{1:d}$ and $\widetilde{w}_{1:d}$, for all $i \in [d]$, we have that $\|w_i - \widetilde{w}_i\| \leq \varepsilon$. Furthermore, by Assumption C.1, we also have that $\|\widetilde{w}_i - w^{z_i}\| \leq \varepsilon/2$. Therefore, by the triangle inequality, we have

$$\left\| w^z - \sum_{i=1}^d \beta_i w^{z_i} \right\| \leq \left\| w^z - \sum_{i=1}^d \beta_i w_i \right\| + \sum_{i=1}^d |\beta_i| \|\widetilde{w}_i - w^{z_i}\| + \sum_{i=1}^d |\beta_i| \|\widetilde{w}_i - w_i\| \leq 3dC\varepsilon/2.$$

This completes the proof. $\qquad\square$

# D   Generic Guarantee for PSDP

In this section, we present a generic guarantee for PSDP (Algorithm 3). We show that given any reward functions $r_{1:h} : \mathcal{X} \times \mathcal{A} \to \mathbb{R}$ and a function class $\mathcal{G} \subseteq \{g : \mathcal{X} \times \mathcal{A} \to \mathbb{R}\}$ that "realizes" these reward functions (we formalize this in the next definition), if $\Psi^{(1:h)}$ are $\alpha$-policy covers for layers 1 through $h$, then for sufficiently large $n \geq 1$ and with high probability, the output $\hat{\pi} = \mathsf{PSDP}(h, r_{1:h}, \mathcal{G}, \Psi^{(1:h)}, n)$ is an approximate maximizer of the objective

$$\max_{\pi \in \Pi_M} \mathbb{E}^\pi \left[ \sum_{t=1}^h r_t(\boldsymbol{x}_t, \boldsymbol{a}_t) \right].$$

To formalize this result, we define the notion of realizability we require for the function class $\mathcal{G}$.

**Definition D.1.** *We say that the function class $\mathcal{G} \subseteq \{g : \mathcal{X} \times \mathcal{A} \to \mathbb{R}\}$ realizes the reward functions $r_{1:h} : \mathcal{X} \times \mathcal{A} \to \mathbb{R}$ if for all $t \in [h]$ and all $\pi \in \Pi_M^{t:h}$,*

$$Q_t^\pi \in \mathcal{G}, \quad \text{where} \qquad Q_t^\pi(x, a) \coloneqq r_t(x, a) + \mathbb{E}^\pi \left[ \sum_{\ell=t+1}^h r_\ell(\boldsymbol{x}_\ell, \boldsymbol{a}_\ell) \,\middle|\, \boldsymbol{x}_t = x, \boldsymbol{a}_t = a \right]. \tag{14}$$

Note that $Q_t^\pi$ in (14) represents the *state-action value function* (Q-function) at layer $t \in [h]$ with respect to the rewards $r_{1:h}$ and partial policy $\pi$.

In what follows, given a function class $\mathcal{G} \subseteq \{g : \mathcal{X} \times \mathcal{A} \to \mathbb{R}\}$, we use $\mathcal{N}_\mathcal{G}(\varepsilon)$ to denote the $\varepsilon$-covering number of $\mathcal{G}$ in $\ell_\infty$ distance. With this, we now state a guarantee for PSDP.

**Theorem D.2.** *Let $\varepsilon, \delta \in (0, 1)$, $B > 0$, and $h \in [H]$. Suppose reward functions $r_{1:h} : \mathcal{X} \times \mathcal{A} \to \mathbb{R}$, function class $\mathcal{G}$, a collection of policies $\Psi^{(1:h)}$, and a parameter $n \geq 1$ satisfy the following:*

- *The function class $\mathcal{G}$ realizes the reward functions $r_{1:h}$ (in the sense of Definition D.1), functions in $\mathcal{G}$ are uniformly bounded by $B$, and $\lim_{n \to \infty} n^{-1} \cdot \log \mathcal{N}_\mathcal{G}(1/n) = 0$.*

- *For some $0 < \alpha \leq 1$, for each $1 \leq t \leq h$, it holds that $\Psi^{(t)}$ is an $\alpha$-policy cover for layer $t$ and moreover $|\Psi^{(t)}| \leq d$.*

- *The parameter $n$ is chosen such that $cdH\alpha^{-1} \cdot \varepsilon_{\mathsf{stat}}(n, \delta/H) \leq \varepsilon$, where $\varepsilon_{\mathsf{stat}}(n, \delta') \coloneqq \sqrt{B^2 n^{-1} \cdot (\log \mathcal{N}_\mathcal{G}(1/n) + \log(n/\delta))}$ and $c > 0$ is a large enough absolute constant.*

*Then, with probability at least $1 - \delta$, the policy $\hat{\pi} = \mathsf{PSDP}(h, r_{1:h}, \mathcal{G}, \Psi^{(1:h)}, n)$ coming from [Algorithm 3](#), satisfies the following guarantee:*

$$\max_{\pi \in \Pi_{\mathsf{M}}} \mathbb{E}^{\pi} \left[ \sum_{t=1}^{h} r_t(\boldsymbol{x}_t, \boldsymbol{a}_t) \right] \leq \mathbb{E}^{\hat{\pi}} \left[ \sum_{t=1}^{h} r_t(\boldsymbol{x}_t, \boldsymbol{a}_t) \right] + \varepsilon.$$

To prove the theorem, we need two intermediate results. The first shows that the $Q$ function is the Bayes-optimal predictor of the sum of rewards when rolling out with policy $\pi$.

**Lemma D.1.** *Let $t \in [H]$, $r_{1:h} : \mathcal{X} \times \mathcal{A} \to \mathbb{R}$ be reward functions, $\pi \in \Pi_{\mathsf{M}}$, and let $g^{\pi}_{\mathsf{bayes}}$ denote the Bayes-optimal predictor[9] for the sum of rewards under policy $\pi$, i.e.,*

$$g^{\pi}_{\mathsf{bayes}} \in \arg\min_{g : \mathcal{X}_t \times \mathcal{A} \to \mathbb{R}} \mathbb{E}^{\pi} \left[ \left( g(\boldsymbol{x}_t, \boldsymbol{a}_t) - \sum_{\ell=t}^{h} r_\ell(\boldsymbol{x}_\ell, \boldsymbol{a}_\ell) \right)^2 \right], \tag{15}$$

*Then, $g^{\pi}_{\mathsf{bayes}} = Q^{\pi}_t$, where $Q^{\pi}_t$ is the $Q$-function at layer $t \in [h]$ with respect to the policy $\pi$ and rewards $r_{1:h}$ defined in [(14)](#).*

**Proof of [Lemma D.1](#).** The least-squares solution $g^{\pi}_{\mathsf{bayes}}$ of the problem in [(15)](#) satisfies, for all $a \in \mathcal{A}$ and $x \in \mathcal{X}_t$,

$$
\begin{aligned}
g^{\pi}_{\mathsf{bayes}}(x, a) &= \mathbb{E}^{\pi} \left[ \sum_{\ell=t}^{h} r_\ell(\boldsymbol{x}_\ell, \boldsymbol{a}_\ell) \,\middle|\, \boldsymbol{x}_t = x, \boldsymbol{a}_t = a \right], \\
&= \mathbb{E}[r_t(\boldsymbol{x}_t, \boldsymbol{a}_t) \mid \boldsymbol{x}_t = x, \boldsymbol{a}_t = a] + \mathbb{E}^{\pi} \left[ \sum_{\ell=t+1}^{h} r_\ell(\boldsymbol{x}_\ell, \boldsymbol{a}_\ell) \,\middle|\, \boldsymbol{x}_t = x, \boldsymbol{a}_t = a \right], \\
&= r_t(x, a) + \mathbb{E}^{\pi} \left[ \sum_{\ell=t+1}^{h} r_\ell(\boldsymbol{x}_\ell, \boldsymbol{a}_\ell) \,\middle|\, \boldsymbol{x}_t = x, \boldsymbol{a}_t = a \right], \\
&= Q^{\pi}_t(s, a),
\end{aligned}
$$

where the last step follows by the definition of the $Q$-function in [(14)](#). $\qquad\square$

We now show that the solution $\hat{g}^{(t)}$ of the least-squares problem in [(6)](#) of [Algorithm 3](#) is close to the $Q$-function in the appropriate sense.

**Lemma D.2.** *Let $\varepsilon, \delta \in (0, 1)$, $B > 0$, and $1 \leq t \leq h \leq H$. Further, let $(\varepsilon_{\mathsf{stat}}, r_{1:h}, \mathcal{G}, \Psi^{(1:h)}, n)$ be as in [Theorem D.2](#). Then, the solution $\hat{g}^{(t)}$ of the least-squares problem in [(6)](#) in [Algorithm 3](#) when calling $\mathsf{PSDP}(h, r_{1:h}, \mathcal{G}, \Psi^{(1:h)}, n)$ satisfies with probability at least $1 - \delta$,*

$$\mathbb{E}^{\mathsf{unif}(\Psi^{(t)})} \left[ \max_{a \in \mathcal{A}} \left( \hat{g}^{(t)}(\boldsymbol{x}_t, a) - Q^{\hat{\pi}^{(t+1)}}_t(\boldsymbol{x}_t, a) \right)^2 \right] \leq c^2 \cdot \varepsilon^2_{\mathsf{stat}}(n, \delta),$$

*where $\hat{\pi}^{(t+1)} \in \Pi_{\mathsf{M}}^{t+1:h}$ is as in [Algorithm 3](#), and $c > 0$ is an absolute constant.*

**Proof of [Lemma D.2](#).** Fix $t \in [h]$ and let $g^{(t)}_{\mathsf{bayes}} = g^{\pi}_{\mathsf{bayes}}$ be as in [Lemma D.1](#) with

$$\pi = \mathsf{unif}(\Psi^{(t)}) \circ_t \pi_{\mathsf{unif}} \circ_{t+1} \hat{\pi}^{(t+1)},$$

and reward functions $r_{1:h}$ as in the lemma's statement. By [Lemma D.1](#), $g^{(t)}_{\mathsf{bayes}}$ is the Bayes-optimal solution of the least-squares problem in [(6)](#) of [Algorithm 3](#). Thus, since $\mathcal{G}$ realizes the reward functions $r_{1:h}$, a standard uniform-convergence guarantee for least-square regression (see e.g. Mhammedi et al. [29, Proposition B.1] with $\boldsymbol{e} = 0$ almost surely) implies that there exists an absolute constant $c > 0$ (independent of $t, h$, and any other problem parameters) such that with probability at least $1 - \delta$,

$$\mathbb{E}^{\mathsf{unif}(\Psi^{(t)})} \left[ \max_{a \in \mathcal{A}} \left( \hat{g}^{(t)}(\boldsymbol{x}_t, a) - g^{(t)}_{\mathsf{bayes}}(\boldsymbol{x}_t, a) \right)^2 \right] \leq c^2 B^2 \cdot \frac{\log \mathcal{N}_{\mathcal{G}}(1/n) + \log(n/\delta)}{n}.$$

The desired result follows by the fact that $g^{(t)}_{\mathsf{bayes}} \equiv Q^{\hat{\pi}^{(t+1)}}_t$; see [Lemma D.1](#). $\qquad\square$

We also require the classical performance difference lemma from Kakade [25].

---

[9]Observe that because the loss is strongly convex, this predictor is unique up to sets of measure zero, justifying our usage of "the" Bayes optimal reward.

**Lemma D.3** (Performance Difference Lemma). *Let $\pi_\star, \pi \in \Pi_M$ be arbitrary, and $Q_t^\pi$ be as defined in (14). Then, for any $h \geq 1$*

$$\mathbb{E}^{\pi_\star}\left[\sum_{t=1}^h r_t(\boldsymbol{x}_t, \boldsymbol{a}_t)\right] - \mathbb{E}^{\pi}\left[\sum_{t=1}^h r_t(\boldsymbol{x}_t, \boldsymbol{a}_t)\right] = \sum_{t=1}^h \mathbb{E}^{\pi_\star}\left[Q_t^\pi(\boldsymbol{x}_t, \pi_\star(\boldsymbol{x}_t)) - Q_t^\pi(\boldsymbol{x}_t, \pi(\boldsymbol{x}_t))\right].$$

Using these results, we now prove Theorem D.2.

**Proof of Theorem D.2.** We first show that for any $t \in [h]$, there is an event $\mathcal{E}_t$ of probability at least $1 - \delta/H$ under which the learned partial policies $\hat{\pi}^{(t)}, \hat{\pi}^{(t+1)}$ are such that

$$\mathbb{E}^{\pi_\star}\left[Q_t^{\hat{\pi}^{(t+1)}}(\boldsymbol{x}_t, \pi_\star(\boldsymbol{x}_t)) - Q_t^{\hat{\pi}^{(t+1)}}(\boldsymbol{x}_t, \hat{\pi}^{(t)}(\boldsymbol{x}_t))\right] \leq \frac{\varepsilon}{H}, \tag{16}$$

where $\pi_\star \in \arg\max_{\pi \in \Pi_M} \mathbb{E}^\pi\left[\sum_{t=1}^h r_t(\boldsymbol{x}_t, \boldsymbol{a}_t)\right]$ is the optimal policy and $Q_t^\pi$ is the $Q$-function defined in (14). Once we establish (16) for all $t \in [h]$, we will apply the performance difference lemma (Lemma D.3) and the union bound to obtain the desired result.

For any $t \in [h]$, we have

$$\mathbb{E}^{\pi_\star}\left[Q_t^{\hat{\pi}^{(t+1)}}(\boldsymbol{x}_t, \pi_\star(\boldsymbol{x}_t)) - Q_t^{\hat{\pi}^{(t+1)}}(\boldsymbol{x}_t, \hat{\pi}^{(t)}(\boldsymbol{x}_t))\right]$$
$$= \mathbb{E}^{\pi_\star}\left[Q_t^{\hat{\pi}^{(t+1)}}(\boldsymbol{x}_t, \pi_\star(\boldsymbol{x}_t)) - \hat{g}^{(t)}(\boldsymbol{x}_t, \pi_\star(\boldsymbol{x}_t)) + \hat{g}^{(t)}(\boldsymbol{x}_t, \pi_\star(\boldsymbol{x}_t)) - Q_t^{\hat{\pi}^{(t+1)}}(\boldsymbol{x}_t, \hat{\pi}^{(t)}(\boldsymbol{x}_t))\right],$$
$$\leq \mathbb{E}^{\pi_\star}\left[Q_t^{\hat{\pi}^{(t+1)}}(\boldsymbol{x}_t, \pi_\star(\boldsymbol{x}_t)) - \hat{g}^{(t)}(\boldsymbol{x}_t, \pi_\star(\boldsymbol{x}_t)) + \hat{g}^{(t)}(\boldsymbol{x}_t, \hat{\pi}^{(t)}(\boldsymbol{x}_t)) - Q_t^{\hat{\pi}^{(t+1)}}(\boldsymbol{x}_t, \hat{\pi}^{(t)}(\boldsymbol{x}_t))\right],$$
$$\leq 2 \cdot \mathbb{E}^{\pi_\star}\left[\max_{a \in \mathcal{A}}\left|Q_t^{\hat{\pi}^{(t+1)}}(\boldsymbol{x}_t, a) - \hat{g}^{(t)}(\boldsymbol{x}_t, a)\right|\right], \tag{17}$$

where the penultimate inequality follows by the fact that $\hat{\pi}^{(t)}(x) \in \arg\max_{a \in \mathcal{A}} \hat{g}^{(t)}(x, a)$, for all $(x, a) \in \mathcal{X}_t \times \mathcal{A}$ by its construction in (8). Now, using the assumption on $\Psi^{(t)}$, we have for any $t \in [t]$,

$$\mathbb{E}^{\pi_\star}\left[Q_t^{\hat{\pi}^{(t+1)}}(\boldsymbol{x}_t, \pi_\star(\boldsymbol{x}_t)) - Q_t^{\hat{\pi}^{(t+1)}}(\boldsymbol{x}_t, \hat{\pi}^{(t)}(\boldsymbol{x}_t))\right]$$
$$\leq 2 \int_{\mathcal{X}_t}\left(\max_{a \in \mathcal{A}}\left|Q_t^{\hat{\pi}^{(t+1)}}(x, a) - \hat{g}^{(t)}(x, a)\right|\right) d^{\pi_\star}(x) \mathrm{d}\nu(x),$$
$$\leq 2\alpha^{-1} \int_{\mathcal{X}_t}\left(\max_{a \in \mathcal{A}}\left|Q_t^{\hat{\pi}^{(t+1)}}(x, a) - \hat{g}^{(t)}(x, a)\right|\right) \max_{\pi \in \Psi^{(t)}} d^\pi(x) \mathrm{d}\nu(x),$$
$$\leq 2\alpha^{-1} d \int_{\mathcal{X}_t}\left(\max_{a \in \mathcal{A}}\left|Q_t^{\hat{\pi}^{(t+1)}}(x, a) - \hat{g}^{(t)}(x, a)\right|\right) d^{\mathrm{unif}(\Psi^{(t)})}(x) \mathrm{d}\nu(x), \quad (|\Psi^{(t)}| \leq d)$$
$$= 2\alpha^{-1} d \cdot \mathbb{E}^{\mathrm{unif}(\Psi^{(t)})}\left[\max_{a \in \mathcal{A}}\left|Q_t^{\hat{\pi}^{(t+1)}}(\boldsymbol{x}_t, a) - \hat{g}^{(t)}(\boldsymbol{x}_t, a)\right|\right], \tag{18}$$

where the first inequality is by (17), the second inequality follows from the fact that $\Psi^{(t)}$ is an $\alpha$-policy cover, the third inequality follows from the fact that $|\Psi^{(t)}| \leq d$, and the equality is by definition. Now, by Lemma D.2, we have that for any $t \in [h]$, there is an absolute constant $c > 0$ (independent of $t$ and other problem parameters) and an event $\mathcal{E}_t$ of probability at least $1 - \delta/H$ under which the solution $\hat{g}^{(t)}$ of the least-squares regression problem on (6) of Algorithm 3 satisfies,

$$\mathbb{E}^{\mathrm{unif}(\Psi^{(t)})}\left[\max_{a \in \mathcal{A}}\left|\hat{g}^{(t)}(\boldsymbol{x}_t, a) - Q_t^{\hat{\pi}^{(t+1)}}(\boldsymbol{x}_t, a)\right|\right] \leq c \cdot \varepsilon_{\mathrm{stat}}(n, \tfrac{\delta}{H}) \leq \frac{\alpha\varepsilon}{2dH}, \tag{19}$$

where the last inequality follows by the choice of $n$ in the theorem's statement. Combining (19) with (18) establishes (16) under the event $\mathcal{E}_t$.

Now, by the performance difference lemma (Lemma D.3), we have

$$\mathbb{E}^{\pi_\star}\left[\sum_{t=1}^h r_t(\boldsymbol{x}_t, \boldsymbol{a}_t)\right] - \mathbb{E}^{\hat{\pi}}\left[\sum_{t=1}^h r_t(\boldsymbol{x}_t, \boldsymbol{a}_t)\right]$$
$$= \sum_{t=1}^h \mathbb{E}^{\pi_\star}\left[Q_t^{\hat{\pi}^{(t+1)}}(\boldsymbol{x}_t, \pi_\star(\boldsymbol{x}_t)) - Q_t^{\hat{\pi}^{(t+1)}}(\boldsymbol{x}_t, \hat{\pi}^{(t)}(\boldsymbol{x}_t))\right].$$

**Algorithm 6** RepLearn($h, \mathcal{F}, \Phi, P, n$): Representation Learning for Low-Rank MDPs [35]

**Require:**

- Target layer $h \in [H]$.
- Discriminator class $\mathcal{F}$.
- Feature class $\Phi$.
- Policy distribution $P \in \Delta(\Pi_{\mathsf{M}})$.
- Number of samples $n \in \mathbb{N}$.

1: Set $\varepsilon_{\mathsf{stat}} = O(\sqrt{cd^2 n^{-1} \log(|\Phi|/\delta)})$ for sufficiently absolute constant $c > 0$ (see Appendix E).
2: Let $\phi^{(1)} \in \Phi$ be arbitrary.
3: Set $\mathcal{D} \leftarrow \varnothing$.
4: **for** $n$ times **do**
5:     Sample $\boldsymbol{\pi} \sim P$.
6:     Sample $(\boldsymbol{x}_h, \boldsymbol{a}_h, \boldsymbol{x}_{h+1}) \sim \boldsymbol{\pi} \circ_h \pi_{\mathsf{unif}}$.
7:     Update dataset: $\mathcal{D} \leftarrow \mathcal{D} \cup \{(\boldsymbol{x}_h, \boldsymbol{a}_h, \boldsymbol{x}_{h+1})\}$.
8: Define $\mathcal{L}_{\mathcal{D}}(\phi, w, f) = \sum_{(x,a,x') \in \mathcal{D}} (\phi(x,a)^\top w - f(x'))^2$.
9: **for** $t = 1, 2, \ldots$ **do**

   /* Discriminator selection */
10:     Solve

$$f^{(t)} \in \arg\max_{f \in \mathcal{F}} \widehat{\Delta}(f), \quad \text{where} \quad \widehat{\Delta}(f) := \max_{\tilde{\phi} \in \Phi} \left\{ \min_{w \in \mathcal{B}(3d^{3/2})} \mathcal{L}_{\mathcal{D}}(\phi^{(t)}, w, f) - \min_{\tilde{w} \in \mathcal{B}(2\sqrt{d})} \mathcal{L}_{\mathcal{D}}(\tilde{\phi}, \tilde{w}, f) \right\}.$$

11:     **if** $\widehat{\Delta}(f^{(t)}) \leq 16 dt \varepsilon_{\mathsf{stat}}^2$ **then**
12:         Return $\phi^{(t)}$.

   /* Feature selection via least-squares minimization */
13:     Solve

$$\phi^{(t+1)} \in \arg\min_{\phi \in \Phi} \min_{(w_1, \ldots, w_t) \in \mathcal{B}(2\sqrt{d})^t} \sum_{\ell=1}^{t} \mathcal{L}_{\mathcal{D}}(\phi, w_\ell, f^{(\ell)}).$$

---

Thus, under the event $\mathcal{E} := \bigcup_{t=1}^{h} \mathcal{E}_t$, we have that

$$\mathbb{E}^{\pi_\star}\left[ \sum_{t=1}^{h} r_t(\boldsymbol{x}_t, \boldsymbol{a}_t) \right] - \mathbb{E}^{\hat{\pi}}\left[ \sum_{t=1}^{h} r_t(\boldsymbol{x}_t, \boldsymbol{a}_t) \right] \leq \varepsilon.$$

The desired result follows by the fact that a union bound implies $\mathbb{P}[\mathcal{E}] \geq 1 - \delta$. $\qquad\square$

## E  Guarantee for RepLearn

In this section, we give a generic guarantee for RepLearn (Algorithm 6). Compared to previous guarantees in Modi et al. [35], Zhang et al. [49], we prove a fast $1/n$-type rate of convergence for RepLearn, and show that the algorithm succeeds even when the norm of the weight $w$ in Line 10 does not grow with the number of iterations. We also use the slightly simpler discriminator class:

$$\mathcal{F} := \left\{ f : x \mapsto \max_{a \in \mathcal{A}} \theta^\top \phi(x, a) \,\middle|\, \theta \in \mathcal{B}(1), \phi \in \Phi \right\}. \tag{20}$$

The main guarantee for RepLearn is as follows.

**Theorem E.1.** *Let $h \in [H]$, $\delta \in (0, e^{-1})$, and $n \in \mathbb{N}$ be given, and suppose that $\mu_{h+1}^{\star}$ satisfies the normalization assumption in [Eq. (4)](#). For any function $f \in \mathcal{F}$, define*

$$w_f = \int_{\mathcal{X}_{h+1}} f(x) \mu_{h+1}^{\star}(x) \mathrm{d}\nu(x).$$

*Let $P \in \Delta(\Pi_{\mathsf{M}})$ be a distribution over policies, $\mathcal{F}$ be as [(20)](#), and $\Phi$ be a feature class satisfying [Assumption 2.2](#). With probability at least $1 - \delta$, RepLearn with input $(h, \mathcal{F}, \Phi, P, n)$ terminates after $t \leq T := \lceil d \log_{3/2}(2nd^{-1/2}) \rceil$ iterations, and its output $\phi^{(t)}$ satisfies*

$$\sup_{f \in \mathcal{F}} \inf_{w \in \mathcal{B}(3d^{3/2})} \mathbb{E}_{\pi \sim P} \mathbb{E}^{\pi \circ_h \pi_{\mathsf{unif}}} \left[ \left( w^{\top} \phi^{(t)}(\boldsymbol{x}_h, \boldsymbol{a}_h) - w_f^{\top} \phi_h^{\star}(\boldsymbol{x}_h, \boldsymbol{a}_h) \right)^2 \right] \leq \varepsilon_{\mathsf{RepLearn}}^2(n, \delta), \qquad (21)$$

*where $\varepsilon_{\mathsf{RepLearn}}^2(n, \delta) := cTd^3 n^{-1} \log(|\Phi|/\delta)$, for some sufficiently large absolute constant $c > 0$.*

To prove the theorem, we need a technical lemma, which follows from Modi et al. [35, Lemma 14].

**Lemma E.1.** *Consider a call to RepLearn$(h, \mathcal{F}, \Phi, P, n)$ ([Algorithm 6](#)) in the setting of [Theorem E.1](#). Further, let $\mathcal{L}_{\mathcal{D}}$ be as in [Algorithm 6](#) and define*

$$(\phi^{(t)}, \widehat{w}_1^{(t)}, \ldots, \widehat{w}_{t-1}^{(t)}) \in \underset{\phi \in \Phi, (w_1, \ldots, w_{t-1}) \in \mathcal{B}(2\sqrt{d})^{t-1}}{\arg\min} \sum_{\ell=1}^{t-1} \mathcal{L}_{\mathcal{D}}(\phi, w_\ell, f^{(\ell)}). \qquad (22)$$

*For any $\delta \in (0, 1)$, there is an event $\mathcal{E}^{(t)}(\delta)$ of probability at least $1 - \delta$ such that under $\mathcal{E}^{(t)}(\delta)$, if [Algorithm 6](#) does not terminate at iteration $t \geq 1$, then for $w^{(\ell)} := w_{f^{(\ell)}}$:*

$$\sum_{\ell=1}^{t-1} \mathbb{E}_{\pi \sim P} \mathbb{E}^{\pi \circ_h \pi_{\mathsf{unif}}} \left[ \left( \phi^{(t)}(\boldsymbol{x}_h, \boldsymbol{a}_h)^{\top} \widehat{w}_\ell^{(t)} - \phi_h^{\star}(\boldsymbol{x}_h, \boldsymbol{a}_h)^{\top} w^{(\ell)} \right)^2 \right] \leq t \varepsilon_{\mathsf{stat}}^2(n, \delta), \qquad (23)$$

$$\inf_{w \in \frac{3}{2} \mathcal{B}(d^{3/2})} \mathbb{E}_{\pi \sim P} \mathbb{E}^{\pi \circ_h \pi_{\mathsf{unif}}} \left[ \left( \phi^{(t)}(\boldsymbol{x}_h, \boldsymbol{a}_h)^{\top} w - \phi_h^{\star}(\boldsymbol{x}_h, \boldsymbol{a}_h)^{\top} w^{(t)} \right)^2 \right] > 8dt \varepsilon_{\mathsf{stat}}^2(n, \delta),$$

*where $\varepsilon_{\mathsf{stat}}^2(n, \delta) := cd^2 n^{-1} \log(|\Phi|/\delta)$ and $c \geq 1$ is a sufficiently large absolute constant.*

With this, we prove [Theorem E.1](#).

**Proof of [Theorem E.1](#).** Let us abbreviate $\varepsilon := \varepsilon_{\mathsf{stat}}(n, \delta)$, with $\varepsilon_{\mathsf{stat}}(n, \delta)$ defined as in [Lemma E.1](#). Further, let $N := 1 + \lceil d \log_{3/2}(2d^{3/2}/\varepsilon) \rceil$, $\delta' := \frac{\delta}{2N}$, and define

$$\tilde{\varepsilon}_{\mathsf{stat}} := \varepsilon_{\mathsf{stat}}(n, \delta'). \qquad (24)$$

Note that $\varepsilon \leq \tilde{\varepsilon}_{\mathsf{stat}}$ and $N - 1 \leq T$, where $T$ is the number of iterations in the theorem statement; the latter inequality follows by the facts that the absolute constant $c$ in [Lemma E.1](#) is at least 1 and $\log(|\Phi|/\delta) \geq 1$. We define an event $\mathcal{E} := \mathcal{E}^{(1)}(\delta') \cap \cdots \cap \mathcal{E}^{(N)}(\delta')$, where $(\mathcal{E}^t(\cdot))_t$ are the success events in [Lemma E.1](#). Note that $\mathbb{P}[\mathcal{E}] \geq 1 - \delta/2$ by the union bound. Throughout this proof, we condition on the event $\mathcal{E}$.

To begin the proof, we define a sequence of vectors $(v_{1:d}^{(\ell)})_{\ell \geq 0}$ in an inductive fashion, with $v_i^{(\ell)} \in \mathbb{R}^d$ for all $i \in [d]$ and $\ell \geq 0$. For $\ell = 0$, we let $v_i^{(0)} = \varepsilon e_i / d$, for all $i \in [d]$. For $\ell \geq 1$, we consider two cases:

- **Case I:** If
  $$\mathcal{J}^{(\ell)} := \left\{ j \in [d] \,\middle|\, |\det(V_{-j}^{(\ell-1)}, w^{(\ell)})| > (1 + C) \cdot |\det(V^{(\ell-1)})| \right\} \neq \varnothing,$$
  where $V^{(\ell-1)} := (v_1^{(\ell-1)}, \ldots, v_d^{(\ell-1)}) \in \mathbb{R}^{d \times d}$ and $w^{(\ell)} := w_{f^{(\ell)}}$, then we let $j := \arg\min_{j' \in \mathcal{J}^{(\ell)}} j'$ and define
  $$v_i^{(\ell)} := \begin{cases} w^{(\ell)}, & \text{if } i = j, \\ v_i^{(\ell-1)}, & \text{otherwise.} \end{cases}$$

- **Case II**: If $\mathcal{J}^{(\ell)} = \varnothing$, we let $v_i^{(\ell)} = v_i^{(\ell-1)}$, for all $i \in [d]$.

We first show that $\mathcal{J}^{(t)} \neq \varnothing$ at any iteration $t \in [N]$ where RepLearn does not terminate. Let $t \in [N]$ be an iteration where the algorithm does not terminate, and suppose that $\mathcal{J}^{(t)} = \varnothing$. This means that

$$\forall j \in [d], \quad |\det(V_{-j}^{(t-1)}, w^{(t)})| \leq (1+C) \cdot |\det(V^{(t-1)})|. \tag{25}$$

Now, since $\det(V^{(t-1)}) \neq 0$ (note that $|\det(V^{(t)})|$ is non-decreasing with $t$), we have that $\mathrm{span}(V^{(t-1)}) = \mathbb{R}^d$. Thus, there exist $\beta_1, \ldots, \beta_d \in \mathbb{R}$ be such that $w^{(t)} = \sum_{i=1}^{d} \beta_i v_i^{(t-1)}$. By the linearity of the determinant and (25), we have

$$\forall j \in [d], \quad (1+C)|\cdot\det(V^{(t-1)})| \geq |\det(V_{-j}^{(t-1)}, w^{(t)})|,$$
$$= \left|\det\left(V_{-j}^{(t-1)}, \sum_{i=1}^{d}\beta_i v_i^{(t-1)}\right)\right|,$$
$$= \left|\sum_{i\in[d]}\beta_i \cdot \det(V_{-j}^{(t-1)}, v_i^{(t-1)})\right|,$$
$$= |\beta_j| \cdot |\det(V^{(t-1)})|.$$

This implies that $|\beta_j| \leq (1+C)$ for all $j \in [d]$. Now, note that by the definition of $(v_i^{(t-1)})$, we have that for any $i \in [d]$ such that $v_i^{(t-1)} \neq \varepsilon e_i/d$, there exists $\ell \in [t-1]$ such that $w^{(\ell)} = v_i^{(t-1)}$. Let

$$\mathcal{I}^{(t)} \coloneqq \{i \in [d] \mid v_i^{(t-1)} \neq \varepsilon e_i/d\},$$

and for any $i \in \mathcal{I}^{(t)}$, let $\ell_i \in [t-1]$ be such that $w^{(\ell_i)} = v_i^{(t-1)}$. Further, define

$$\widetilde{w}^{(t)} \coloneqq \sum_{i\in\mathcal{I}^{(t)}}\beta_i w^{(\ell_i)} = \sum_{i\in\mathcal{I}^{(t)}}\beta_i v_i^{(t-1)}, \tag{26}$$

and note that by the triangle inequality and the fact that $w^{(t)} = \sum_{i=1}^{d}\beta_i v_i^{(t-1)}$, we have

$$\|\widetilde{w}^{(t)} - w^{(t)}\| \leq (1+C)\varepsilon_{\mathsf{stat}}. \tag{27}$$

Finally, with the notation in (22), define

$$\widehat{w}_t^{(t)} \coloneqq \sum_{i\in\mathcal{I}^{(t)}}\beta_i \widehat{w}_{\ell_i}^{(t)}, \tag{28}$$

and note that

$$\widehat{w}_t^{(t)} \in (1+C)\mathcal{B}(2d^{3/2}),$$

since $|\beta_i| \leq (1+C)$ for all $i \in [d]$, $|\mathcal{I}^{(t)}| \leq d$, and $\widehat{w}_\ell^{(t)} \in \mathcal{B}(2\sqrt{d})$, for all $\ell \in [t-1]$. Now, by Lemma E.1, in particular (23), we have

$$\sum_{i\in\mathcal{I}^{(t)}}\mathbb{E}_{\pi\sim P}\mathbb{E}^{\pi\circ_h\pi_{\mathsf{unif}}}\left[\left(\phi^{(t)}(\boldsymbol{x}_h,\boldsymbol{a}_h)^{\top}\widehat{w}_{\ell_i}^{(t)} - \phi_h^{\star}(\boldsymbol{x}_h,\boldsymbol{a}_h)^{\top}w^{(\ell_i)}\right)^2\right] \leq t\tilde{\varepsilon}_{\mathsf{stat}}^2, \tag{29}$$

where $\tilde{\varepsilon}_{\mathsf{stat}}$ is as in (24). Using the expressions in Eqs. (26) and (28) with (29) and Jensen's inequality, we have that under $\mathcal{E}^{(t)}$,

$$\mathbb{E}_{\pi\sim P}\mathbb{E}^{\pi\circ_h\pi_{\mathsf{unif}}}\left[\left(\phi^{(t)}(\boldsymbol{x}_h,\boldsymbol{a}_h)^{\top}\widehat{w}_t^{(t)} - \phi_h^{\star}(\boldsymbol{x}_h,\boldsymbol{a}_h)^{\top}\widetilde{w}^{(t)}\right)^2\right]$$
$$\leq \left(\sum_{j\in\mathcal{I}^{(t)}}|\beta_j|\right)\cdot\sum_{i\in\mathcal{I}^{(t)}}\mathbb{E}_{\pi\sim P}\mathbb{E}^{\pi\circ_h\pi_{\mathsf{unif}}}\left[\left(\phi^{(t)}(\boldsymbol{x}_h,\boldsymbol{a}_h)^{\top}\widehat{w}_{\ell_i}^{(t)} - \phi_h^{\star}(\boldsymbol{x}_h,\boldsymbol{a}_h)^{\top}w^{(\ell_i)}\right)^2\right],$$
$$\leq (1+C)dt\tilde{\varepsilon}_{\mathsf{stat}}^2.$$

Now, using (27) and the facts that $(a+b)^2 \leq 2a^2 + 2b^2$ and $\|\phi_h^{\star}\|_2 \leq 1$, we have that

$$\mathbb{E}_{\pi\sim P}\mathbb{E}^{\pi\circ_h\pi_{\mathsf{unif}}}\left[\left(\phi^{(t)}(\boldsymbol{x}_h,\boldsymbol{a}_h)^{\top}\widehat{w}_t^{(t)} - \phi_h^{\star}(\boldsymbol{x}_h,\boldsymbol{a}_h)^{\top}w^{(t)}\right)^2\right] \leq 2(1+C)^2\varepsilon^2 + 2(1+C)dt\tilde{\varepsilon}_{\mathsf{stat}}^2,$$
$$\leq 2(1+C)^2\tilde{\varepsilon}_{\mathsf{stat}}^2 + 2(1+C)dt\tilde{\varepsilon}_{\mathsf{stat}}^2.$$

Using that $C = 1/2$, we conclude that the right-hand side of this inequality is bounded by $8dt\tilde{\varepsilon}^2_{\text{stat}}$ which is a contradiction, since $\widehat{w}^{(t)}_t \in (1+C)\mathcal{B}(2d^{3/2}) = \mathcal{B}(3d^{3/2})$ and by Lemma E.1, we must have

$$\inf_{w \in \mathcal{B}(3d^{3/2})} \mathbb{E}_{\pi \sim P} \mathbb{E}^{\pi \circ_h \pi_{\text{unif}}} \left[ \left( \phi^{(t)}(\boldsymbol{x}_h, \boldsymbol{a}_h)^\top w - \phi^\star_h(\boldsymbol{x}_h, \boldsymbol{a}_h)^\top w^{(t)} \right)^2 \right] > 8t\tilde{\varepsilon}_{\text{stat}}$$

if RepLearn does not terminate at round $t$. Therefore, we have that $\mathcal{J}^{(t)} \neq \varnothing$, for any iteration $t \in [2 .. N]$ where RepLearn does not terminate.

We now bound the iteration count and prove that the guarantee in Eq. (21) holds at termination. Note that whenever $\mathcal{J}^{(\ell)} \neq \varnothing$ for $\ell > 1$, we have by construction:

$$|\det(V^{(\ell)})| > 3/2 \cdot |\det(V^{(\ell-1)})|.$$

Thus, if RepLearn runs for $t \in [2 .. N]$ iterations, then

$$|\det(V^{(t)})| > (3/2)^{t-1} \cdot |\det(V^{(1)})|. \tag{30}$$

On the other hand, since the determinant of a matrix is bounded by the product of the norms of its columns and $v^{(t)}_{1:d} \in \mathcal{B}(2\sqrt{d})$, we have

$$|\det(V^{(t)})| \leq 2^d d^{d/2}.$$

Note also that $|\det(V^{(0)})| = (\varepsilon/d)^d$. Plugging this into (30), we conclude that

$$(3/2)^{t-1} < (2d^{3/2}/\varepsilon)^d.$$

Taking the logarithm on both sides and rearranging yields

$$t < 1 + d\log_{3/2}(2d^{3/2}/\varepsilon) \leq N.$$

Thus, the algorithm must terminate after at most $N - 1$ iterations. Furthermore, by [35, Lemma 14], we have that with probability at least $1 - \frac{\delta}{2N}$, if the algorithm terminates at iteration $t$, then

$$\max_{f \in \mathcal{F}} \inf_{w \in \mathcal{B}(3d^{3/2})} \mathbb{E}_{\pi \sim P} \mathbb{E}^{\pi \circ_h \pi_{\text{unif}}} \left[ \left( w^\top \phi^{(t)}(\boldsymbol{x}_h, \boldsymbol{a}_h) - w_f^\top \phi^\star_h(\boldsymbol{x}_h, \boldsymbol{a}_h) \right)^2 \right] \leq 32t\tilde{\varepsilon}^2_{\text{stat}},$$
$$\leq 32(N-1)\tilde{\varepsilon}^2_{\text{stat}},$$
$$\leq 32T\tilde{\varepsilon}^2_{\text{stat}}.$$

Applying a union bound completes the proof. $\qquad\square$

# F    Analysis

In this section, we prove the main guarantee for SpanRL (Theorem 3.2). First, we outline our proof strategy in Appendix F.1. Then, in Appendix F.2 and Appendix F.3, we present guarantees for the instances of PSDP (Algorithm 3) and RobustSpanner (Algorithm 2) used within SpanRL. We then combine these results in Appendix F.5 to complete the proof of Theorem 3.2. A self-contained guarantee for RobustSpanner(Lemma 3.1) is given in Appendix F.6.

## F.1    Proof Strategy

Like our algorithm, our analysis is inductive. For fixed $h$, we assume that the policy set $\Psi^{(1:h+1)}$ produced by SpanRL satisfies the property:

$$\Psi^{(1)}, \ldots \Psi^{(h+1)} \text{ are } \left( \tfrac{1}{4Ad}, 0 \right)\text{-policy covers for layers } 1 \text{ through } h + 1, \text{ and } \max_{t \in [h+1]} |\Psi^{(t)}| \leq d. \tag{31}$$

Conditioned on this claim, we show that with high probability, the set $\Psi^{(h+2)}$ is a $\left( \tfrac{1}{4Ad}, 0 \right)$-policy cover for layer $h + 2$. To prove this, we use the inductive assumption to show that PSDP acts as an approximate linear optimization oracle over $\mathcal{W} = \left\{ \mathbb{E}^\pi \left[ \phi^{(h)}(\boldsymbol{x}_h, \boldsymbol{a}_h) \right] \mid \pi \in \Pi_{\mathsf{M}} \right\}$ (Appendix F.2). Using this, we then instantiate the guarantee of RobustSpanner from Lemma F.3 with LinOpt and LinEst instantiated with PSDP and EstVec. To conclude the proof of the inductive step, we the main guarantee for RobustSpanner together with the main guarantee for RepLearn (Theorem E.1), along with a change of measure argument enabled by the assumption that $\Psi^{(1:h)}$ are policy covers (i.e. (31)).

## F.2 Guarantee for PSDP as a Subroutine for `RobustSpanner`

We begin by showing that PSDP instantiates the approximate linear optimization oracle required by `RobustSpanner`. In particular, we fix a layer $h$ and assume that $\Psi^{(1:h+1)}$ satisfy (31) and apply the results of Appendix D.

More precisely, we need to show that, for any $\theta \in \mathbb{R}^d \smallsetminus \{0\}$ and $\phi \in \Phi$, PSDP approximately solves

$$\max_{\pi \in \Pi_{\mathsf{M}}} \theta^\top \mathbb{E}^\pi [\phi(\boldsymbol{x}_h, \boldsymbol{a}_h)]. \tag{32}$$

We can equivalently formulate (32) as, for fixed $\theta \in \mathbb{R}^d \smallsetminus \{0\}$ and $\phi \in \Phi$, maximizing the sum of the reward functions $r_{1:h}(\cdot, \cdot; \theta, \phi)$ given by:

$$\forall (x, a) \in \mathcal{X} \times \mathcal{A}, \quad r_t(x, a; \theta, \phi) := \begin{cases} \phi(x, a)^\top \frac{\theta}{\|\theta\|}, & \text{for } t = h, \\ 0, & \text{otherwise.} \end{cases} \tag{33}$$

Note that this matches the choice of reward functions in `SpanRL` (Algorithm 1) at iteration $h$ with $\phi = \hat{\phi}^{(h)}$, the feature map returned by `RepLearn` in Line 8. With these reward functions and the function class

$$\mathcal{G} := \{g : (x, a) \mapsto \phi(x, a)^\top w \mid \phi \in \Phi, w \in \mathcal{B}(\sqrt{d})\}, \tag{34}$$

we show that the output $\hat{\pi} = \mathsf{PSDP}(h, r_{1:h}(\cdot, \cdot; \theta, \phi), \mathcal{G}, \Psi^{(1:h)}, n)$ approximately solves (32) with high probability if $n \geq 1$ is sufficiently large. We first verify that the class $\mathcal{G}$ realizes the reward functions specified in (33) in the sense of Definition D.1.

**Lemma F.1.** *Under Assumption 2.2, the function class $\mathcal{G}$ in (34) realizes the reward functions in (33), for any $\phi \in \Phi$ and $\theta \in \mathbb{R}^d \smallsetminus \{0\}$. Furthermore, we have that functions in $\mathcal{G}$ are uniformly bounded by $h \leq H$, and $\log \mathcal{N}_{\mathcal{G}}(\varepsilon) \leq \log |\Phi| + d \log(H/\varepsilon)$, where we recall that $\mathcal{N}_{\mathcal{G}}(\varepsilon)$ denotes the $\varepsilon$-covering number of $\mathcal{G}$ in $\ell_\infty$ distance.*

**Proof.** Fix $\phi \in \Phi$ and $\theta \in \mathbb{R}^d \smallsetminus \{0\}$, and let $r_\ell(\cdot, \cdot) \equiv r_\ell(\cdot, \cdot; \theta, \phi)$, for $\ell \in [h]$. For $t = h$, we clearly have that for any $\pi \in \Pi_{\mathsf{M}}^{h:h}$, $Q_t^\pi(\cdot, \cdot) = r_t(\cdot, \cdot) \in \mathcal{G}$. For $t < h$ and $\pi \in \Pi_{\mathsf{M}}^{t:h}$, we have by the low-rank structure that

$$Q_t^\pi(x, a) = \int_{\mathcal{X}_{t+1}} \mathbb{E}^\pi [r_h(\boldsymbol{x}_h, \boldsymbol{a}_h) \mid \boldsymbol{x}_{t+1} = y, \boldsymbol{a}_{t+1} = \pi(y)] \cdot \phi_t^\star(x, a)^\top \mu_{t+1}^\star(y) \mathrm{d}\nu(y),$$

$$= \phi_t^\star(x, a)^\top \left( \int_{\mathcal{X}_{t+1}} \mathbb{E}^\pi [r_h(\boldsymbol{x}_h, \boldsymbol{a}_h) \mid \boldsymbol{x}_{t+1} = y, \boldsymbol{a}_{t+1} = \pi(y)] \cdot \mu_{t+1}^\star(y) \mathrm{d}\nu(y) \right). \tag{35}$$

Now, by the fact that $\mathbb{E}^\pi [r_h(\boldsymbol{x}_h, \boldsymbol{a}_h) \mid \boldsymbol{x}_{t+1} = y, \boldsymbol{a}_{t+1} = \pi(y)] \in [-1, 1]$, for all $y \in \mathcal{X}_{t+1}$ (since $\phi(\cdot, \cdot) \in \mathcal{B}(1)$, for all $\phi \in \Phi$, and the normalizing assumption made on $(\mu_h^\star)_{h \in [H]}$ in Section 2.2 (i.e. that for all $g : \mathcal{X}_{t+1} \to [0, 1]$, $\left\| \int_{\mathcal{X}_{t+1}} \mu_{t+1}^\star(y) g(y) \mathrm{d}\nu(y) \right\| \leq \sqrt{d}$), we have that

$$w_t := \int_{\mathcal{X}_{t+1}} \mathbb{E}^\pi [r_h(\boldsymbol{x}_h, \boldsymbol{a}_h) \mid \boldsymbol{x}_{t+1} = y, \boldsymbol{a}_{t+1} = \pi(y)] \cdot \mu_{t+1}^\star(y) \mathrm{d}\nu(y) \in \mathcal{B}(\sqrt{d}).$$

This together with (35) and the fact that $\phi_t^\star \in \Phi$ (by Assumption 2.2), we have that $Q_t^\pi \in \mathcal{G}$. $\qquad \square$

Combining Lemma F.1 with Theorem D.2 results in the following bound on the quality of PSDP as an approximate linear optimization oracle over the space of policies.

**Corollary F.1.** *Let $\varepsilon, \delta \in (0, 1)$ and $h \in [H]$. Further, let $\theta \in \mathbb{R}^d \smallsetminus \{0\}$, $\phi \in \Phi$, and $\hat{\pi} = \mathsf{PSDP}(h, r_{1:h}(\cdot, \cdot; \theta, \phi), \mathcal{G}, \Psi^{(1:h)}, n)$, where*

- *The reward functions $r_{1:h}(\cdot, \cdot; \theta, \phi)$ are as in (33).*

- *The function class $\mathcal{G}$ is as in (34).*

- *The collection of policies $\Psi^{(1:h)}$ satisfy (31).*

- *The parameter $n$ is chosen such that $cHACd^2 \cdot \varepsilon_{\mathsf{stat}}(n, \delta/H) \leq \varepsilon$, where $\varepsilon_{\mathsf{stat}}(n, \delta') := \sqrt{dn^{-1} \cdot (d \log(nH) + \log(|\Phi|/\delta'))}$ and $c > 0$ is some large enough absolute constant.*

*Then, under Assumption 2.2, with probability at least $1 - \delta$, we have that*

$$\max_{\pi \in \Pi_{\mathsf{M}}} \theta^\top \mathbb{E}^\pi [\phi(\boldsymbol{x}_h, \boldsymbol{a}_h)] \leq \theta^\top \mathbb{E}^{\hat{\pi}} [\phi(\boldsymbol{x}_h, \boldsymbol{a}_h)] + \varepsilon/2.$$

We emphasize that the inductive assumption that $\Psi^{(1:h)}$ is a policy cover of bounded size enters only in the statement of Theorem D.2. We now give a guarantee for `RobustSpanner` as used in `SpanRL`.

---

**Algorithm 7** EstVec($h, F, \pi, n$): Estimate $\mathbb{E}^\pi[F(\boldsymbol{x}_h, \boldsymbol{a}_h)]$ for given policy $\pi$ and function $F$.

**Require:**

- Target layer $h \in [H]$.
- Vector-valued function $F : \mathcal{X} \times \mathcal{A} \to \mathbb{R}^d$.
- Policy $\pi \in \Pi_\mathsf{M}$.
- Number of samples $n \in \mathbb{N}$.

1: $\mathcal{D} \leftarrow \varnothing$.
2: **for** $n$ times **do**
3:     Sample $(\boldsymbol{x}_h, \boldsymbol{a}_h) \sim \pi$.
4:     Update dataset: $\mathcal{D} \leftarrow \mathcal{D} \cup \{(\boldsymbol{x}_h, \boldsymbol{a}_h)\}$.
5: **Return:** $\bar{F} = \frac{1}{n} \sum_{(x,a) \in \mathcal{D}} F(x, a)$.

---

### F.3 Guarantee for RobustSpanner as a Subroutine for SpanRL

In this section, we prove a guarantee for the instantiation of RobustSpanner in SpanRL, which we require in the proof of the main theorem (Theorem 3.2). We first show that the LinEst subroutine passed to RobustSpanner can be taken to be EstVec (Algorithm 7), which simply estimates the expected feature imbedding of $(\boldsymbol{x}_h, \boldsymbol{a}_h)$ under policy $\pi$ by sampling sufficiently many trajectories and taking the empirical mean.

**Lemma F.2** (Guarantee of EstVec). *Let $\delta \in (0, 1)$ and $\varepsilon > 0$. For $h \in [H]$, $\phi \in \Phi$, $\pi \in \Pi_\mathsf{M}$, and $n \in \mathbb{N}$ such that $n \geq \frac{c}{\varepsilon^2} \log(d/\delta)$ for some large enough absolute constant $c > 0$, the output $\bar{\phi}_h = \mathsf{EstVec}(h, \phi, \pi, n)$ (Algorithm 7) satisfies, with probability at least $1 - \delta$,*

$$\|\bar{\phi}_h - \mathbb{E}^\pi[\phi(\boldsymbol{x}_h, \boldsymbol{a}_h)]\| \leq \varepsilon/2.$$

**Proof.** By Hoeffding's inequality (see for example [22, Corollary 7]) and the fact that $\|\phi(x, a)\| \leq 1$ for all $x \in \mathcal{X}$ and $a \in \mathcal{A}$, there exists an absolute constant $c > 0$ such that with probability at least $1 - \delta$,

$$\|\bar{\phi}_h - \mathbb{E}^\pi[\phi(\boldsymbol{x}_h, \boldsymbol{a}_h)]\| \leq c \cdot \sqrt{\frac{\log(d/\delta)}{n}}.$$

Setting $n$ as in the statement of the theorem concludes the proof. $\qquad\square$

In SpanRL, we instantiate RobustSpanner passing PSDP as LinOpt and EstVec as LinEst. Combining Corollary F.1 and Lemma F.2 with the general guarantee of RobustSpanner in Proposition C.1, we have the following result.

**Lemma F.3.** *Consider iteration $h \in [H]$ of $\mathsf{SpanRL}(\Phi, \varepsilon, \mathfrak{c}, \delta)$ (Algorithm 1) with $\varepsilon, \mathfrak{c} > 0$, $\delta \in (0, 1)$, and feature class $\Phi$ satisfying Assumption 2.2. Further, let $\hat{\phi}^{(h)}$ denote the feature map returned by RepLearn in Algorithm 1 at iteration $h$. If $\Psi^{(1:h)}$ satisfy (31) and $\mathfrak{c} = \mathrm{polylog}(A, d, H, \log(|\Phi|/\delta))$ is large enough, then there is an event $\mathcal{E}_h$ with probability at least $1 - \frac{\delta}{2H}$ such that*

- *The number of iterations of RobustSpanner in Line 12 of Algorithm 1 is at most $N = \left\lceil \frac{d}{2} \log_2 \left( \frac{100d}{\varepsilon} \right) \right\rceil < \infty$, and*

- *The output $(\pi_1, \ldots, \pi_d)$ of RobustSpanner has the property that for all $\pi \in \Pi_\mathsf{M}$, there exist $\beta_1, \ldots, \beta_d \in [-2, 2]$ such that*

$$\left\| \hat{\phi}^{(h),\pi} - \sum_{i=1}^d \beta_i \hat{\phi}^{(h),\pi_i} \right\| \leq 3d\varepsilon, \quad \text{where} \quad \hat{\phi}^{(h),\pi'} \coloneqq \mathbb{E}^{\pi'}\left[\hat{\phi}^{(h)}(\boldsymbol{x}_h, \boldsymbol{a}_h)\right].$$

**Proof.** By Proposition C.1, on the event that the instances of PSDP and EstVec used by RobustSpanner satisfy Assumption C.1 with $\varepsilon' = \frac{\varepsilon}{2}$, the two desiderata of the lemma hold.[10] We

---

[10]Here, we instantiate the guarantee in Proposition C.1 with $C = 2$; this is what $C$ is set to in Algorithm 1.

claim that each call to PSDP and to EstVec satisfies Assumption C.1 with probability at least $1 - \frac{\delta}{8dNH}$. Because each of PSDP and EstVec get called at most $4dN$ times per iteration of RobustSpanner, a union bound concludes the proof contingent on the above claim.

We now prove the claim. First, note that the instance of PSDP that RobustSpanner uses within Algorithm 1 is of the form:

$$\mathsf{PSDP}(h, r_{1:h}(\cdot, \cdot, \theta), \mathcal{G}, \Psi^{1:h}, n_{\mathsf{PSDP}})$$

with $r_{1:h}$ and $\mathcal{G}$ as in Algorithm 1; this matches the form in Corollary F.1 (PSDP's guarantee) with $\phi = \hat{\phi}^{(h)}$. Thus, by choosing

$$n_{\mathsf{PSDP}} = \mathfrak{c} \cdot \frac{A^2 d^5 H^2 \cdot (d\log(H) + \log(8dH^2 N |\Phi|/\delta))}{\varepsilon^2},$$

for $\mathfrak{c} = \operatorname{polylog}(A, d, H, \log(|\Phi|/\delta))$ sufficiently large, the conditions of Corollary F.1 are satisfied, and its conclusion implies the claim for the PSDP instance used by RobustSpanner. Similarly, the choice of $n_{\mathsf{EstVec}}$ in Algorithm 1 ensures that the claim holds for the instance of EstVec that RobustSpanner uses by Lemma F.2. The result follows.

$\square$

### F.4 Guarantee for RepLearn as a Subroutine for SpanRL

In this section, we prove a guarantee for the invocation of RepLearn within SpanRL

Recall that $P^{(h)} = \mathsf{unif}(\Psi^{(h)})$ is the distribution over policies that SpanRL passes to RepLearn at iteration $h \in [H-2]$ to compute feature map $\phi^{(h)}$. Thus, by invoking Theorem E.1 in Appendix E and using the choice of $n_{\mathsf{RepLearn}}$ in Algorithm 1, we immediately obtain the following corollary.

**Corollary F.2.** *Let $\delta, \varepsilon \in (0,1)$, and $\mathcal{F}$ be as in Algorithm 1, and fix $h \in [H-2]$. Suppose that the feature class $\Phi$ satisfies Assumption 2.2. Then, with probability at least $1 - \frac{\delta}{2H}$, the instance of RepLearn in Line 9 of Algorithm 1 runs for $t \le \mathfrak{c} \cdot d$ iterations for $\mathfrak{c} = \operatorname{polylog}(A, d, H, \log(|\Phi|/\delta))$ sufficiently large, and returns output $\phi^{(h)}$ such that for all $f \in \mathcal{F}$, there exists $w_f^{(h)} \in \mathcal{B}(3d^{3/2})$ satisfying*

$$\mathbb{E}^{\mathsf{unif}(\Psi^{(h)})} \left[ \sum_{a \in \mathcal{A}} \left( \phi^{(h)}(\boldsymbol{x}_h, a)^\top w_f^{(h)} - \phi_h^\star(\boldsymbol{x}_h, a)^\top w_f \right)^2 \right] \le \frac{\eta^2}{64 A^2 d^2},$$

*where $w_f \coloneqq \int_{\mathcal{X}_{h+1}} f(y) \mu_{h+1}^\star(y) \mathrm{d}\nu(y)$.*

### F.5 Concluding the Proof of Theorem 3.2

In this section, we conclude the proof of the main guarantee (Theorem 3.2). We derive the guarantee from the following inductive claim.

**Theorem F.1.** *Consider iteration $h \in [H]$ of $\mathsf{SpanRL}(\Phi, \varepsilon, \mathfrak{c}, \delta)$ (Algorithm 1) with parameters $\varepsilon, \mathfrak{c} > 0$, $\delta \in (0,1)$ and a feature class $\Phi$ satisfying Assumption 2.2. Further, assume that:*

- *The collection of policies $\Psi^{(1:h+1)}$ at the start of the $h$th iteration of SpanRL satisfy (31).*

- *Assumption 2.1 (reachability) holds with $\eta > 0$.*

- *The input parameter $\varepsilon$ to SpanRL is set to $\varepsilon = \frac{\eta}{36d^{5/2}}$.*

- *The input parameter $\mathfrak{c} = \operatorname{polylog}(A, d, H, \log(|\Phi|/\delta))$ is sufficiently large.*

*Then, with probability at least $1 - \frac{\delta}{H}$, the set of policies $\Psi^{(h+2)}$ produced by $\mathsf{SpanRL}(\Phi, \varepsilon, \mathfrak{c}, \delta)$ at the end of iteration $h$ is an $(\frac{1}{4Ad}, 0)$-policy cover for layer $h + 2$.*

With this, we can now prove Theorem 3.2.

**Proof of Theorem 3.2.** Note that it suffices to prove that (31) holds for $h = H - 1$ with probability at least $1 - \delta$. To do this, we proceed by induction over $h = 1, \ldots, H - 1$. The base case of $h = 1$ trivially holds because $\Psi^{(1)} = \varnothing$ and $\Psi^{(2)} = \{\pi_{\mathsf{unif}}\}$. The induction step now follows by Theorem F.1 and the union bound (see e.g. [30, Lemma I.2]).

The number of trajectories used by SpanRL is dominated by calls to PSDP. Since PSDP is called $O(d \log(d/\varepsilon))$ times at each iteration of SpanRL (Lemma F.3), and each call to PSDP requires at most $Hn_{\mathsf{PSDP}}$ trajectories, the total number of trajectories after $H$ iterations of SpanRL is bounded by $\widetilde{O}(H^2 d n_{\mathsf{PSDP}})$. By plugging the choices for $n_{\mathsf{PSDP}}$ and $\varepsilon$ from the theorem statement, we obtain the claimed sample complexity. $\qquad\square$

Before proving Theorem F.1, we make the following simple observation.

**Lemma F.4.** *For any $\pi \in \Pi_{\mathsf{M}}$, $h \in [H-1]$, any $x \in \mathcal{X}_{h+1}$, we have*
$$\mu_{h+1}^\star(x)^\top \mathbb{E}^\pi[\phi_h^\star(\boldsymbol{x}_h, \boldsymbol{a}_h)] = d^\pi(x) \geq 0.$$

**Proof of Lemma F.4.** The equality follows by construction. The non-negativity of $d^\pi(x)$ follows by definition of a probability density. $\qquad\square$

We now prove Theorem F.1.

**Proof of Theorem F.1.** Let $\mathcal{E}_h$ and $\mathcal{E}_h'$ denote the success events in Lemma F.3 and Corollary F.2, respectively, and note that by the union bound, we have $\mathbb{P}[\mathcal{E}_h \cap \mathcal{E}_h'] \geq 1 - \delta/H$. For the rest of this proof, we will condition on $\mathcal{E} \coloneqq \mathcal{E}_h \cap \mathcal{E}_h'$.

Throughout, we denote
$$\phi_h^{\star,\pi} \coloneqq \mathbb{E}^\pi[\phi_h^\star(\boldsymbol{x}_h, \boldsymbol{a}_h)], \quad \forall h \in [H], \forall \pi \in \Pi_{\mathsf{M}}.$$

Because $\Psi^{(1:h+1)}$ satisfy (31) (i.e., are a policy cover) it holds by Lemma F.4 that for all $x \in \mathcal{X}_h$,
$$\max_{\pi \in \Psi^{(h)}} \mu_h^\star(x)^\top \phi_{h-1}^{\star,\pi} \geq \alpha \cdot \sup_{\pi \in \Pi_{\mathsf{M}}} \mu_h^\star(x)^\top \phi_{h-1}^{\star,\pi}, \quad \text{for} \quad \alpha \coloneqq \frac{1}{4Ad}. \tag{36}$$

We will show that with probability at least $1 - \frac{\delta}{H}$, the policy set $\Psi^{(h+2)}$ has the same property for layer $h+2$; that is, for all $x \in \mathcal{X}_{h+1}$,
$$\max_{\pi \in \Psi^{(h+2)}} \mu_{h+2}^\star(x)^\top \phi_{h+1}^{\star,\pi} \geq \alpha \cdot \sup_{\pi \in \Pi_{\mathsf{M}}} \mu_{h+2}^\star(x)^\top \phi_{h+1}^{\star,\pi}. \tag{37}$$

Again, by Lemma F.4 this is equivalent to the statement that $\Psi^{(h+2)}$ is an $(\frac{1}{4Ad}, 0)$-policy cover for layer $h+2$.

For the remainder of the proof, we will fix $x \in \mathcal{X}_{h+2}$ and let $\pi_x \in \arg\max_{\pi \in \Pi_{\mathsf{M}}} \mu_{h+2}^\star(x)^\top \phi_{h+1}^{\star,\pi}$. Our goal is to show that the inequality Eq. (37) holds for $x$.

**Preliminaries.** Note that since $x \in \mathcal{X}_{h+2}$, we have $\|\mu_{h+2}^\star(x)\| > 0$. It will be convenient to introduce a function $f : \mathcal{X}_{h+1} \to \mathbb{R}$ defined by
$$f(y) \coloneqq \theta_x^\top \phi_{h+1}^\star(y, \pi_x(y)), \quad \text{where} \quad \theta_x \coloneqq \frac{\mu_{h+2}^\star(x)}{\|\mu_{h+2}^\star(x)\|}.$$

Further, we define
$$w_x \coloneqq \int_{\mathcal{X}_{h+1}} f(y) \mu_{h+1}^\star(y) \mathrm{d}\nu(y). \tag{38}$$

By definition of $\pi_x$, we have that for all $y \in \mathcal{X}_{h+1}$,
$$\theta_x^\top \phi_{h+1}^\star(y, \pi_x(y)) = \max_{a \in \mathcal{A}} \theta_x^\top \phi_{h+1}^\star(y, a).$$

This together with the fact that $\|\theta_x\| = 1$ implies that
$$f \in \mathcal{F} = \left\{ x \mapsto \max_{a \in \mathcal{A}} \theta^\top \phi(x, a) \,\middle|\, \theta \in \mathcal{B}(1), \phi \in \Phi \right\}; \tag{39}$$

the discriminator class in Line 4 of SpanRL. Note also that since $x \in \mathcal{X}_{h+2}$, we have by reachability that
$$w_x^\top \phi_h^{\star,\pi_x} = \theta_x^\top \phi_{h+1}^{\star,\pi_x} = \frac{1}{\|\mu_{h+2}^\star(x)\|} \max_{\pi \in \Pi_{\mathsf{M}}} \mu_{h+2}^\star(x)^\top \phi_{h+1}^{\star,\pi} \geq \eta > 0. \tag{40}$$

**Applying the guarantee for RepLearn.** Moving forward, let $\phi^{(h)}$ be the feature map returned by RepLearn at the $h$th iteration of Algorithm 1, and define $\phi^{(h),\pi} \coloneqq \mathbb{E}^\pi[\phi^{(h)}(\boldsymbol{x}_h, \boldsymbol{a}_h)]$, for any $\pi \in \Pi_\mathsf{M}$. Further, let $w_x^{(h)}$ be the vector $w_f^{(h)}$ in Corollary F.2 with $f = f_x$, and note that

$$\|w_x^{(h)}\| \le 3d^{3/2}. \tag{41}$$

By Jensen's inequality, we compute

$$\left(\langle w_x^{(h)}\rangle\phi^{(h),\pi_x} - \langle w_x\rangle\phi_h^{\star,\pi_x}\right)^2$$

$$\le \mathbb{E}^{\pi_x}\left[\left(\phi^{(h)}(\boldsymbol{x}_h, \boldsymbol{a}_h)^\top w_x^{(h)} - \phi_h^\star(\boldsymbol{x}_h, \boldsymbol{a}_h)^\top w_x\right)^2\right], \quad \text{(Jensen's inequality)}$$

$$= \int_{\mathcal{X}_h}\left(\phi^{(h)}(y, \pi_x(y))^\top w_x^{(h)} - \phi_h^\star(y, \pi_x(y))^\top w_x\right)^2 \mu_h^\star(y)^\top \phi_{h-1}^{\star,\pi_x}\mathrm{d}\nu(y), \quad \text{(Low-Rank MDP)}$$

$$\le \alpha^{-1}\max_{\tilde\pi\in\Psi^{(h)}}\int_{\mathcal{X}_h}\left(\phi^{(h)}(y, \pi_x(y))^\top w_x^{(h)} - \phi_h^\star(y, \pi_x(y))^\top w_x\right)^2 \mu_h^\star(y)^\top \phi_{h-1}^{\star,\tilde\pi}\mathrm{d}\nu(y), \quad \text{(by (36))}$$

$$\le \alpha^{-1}\sum_{\tilde\pi\in\Psi^{(h)}}\int_{\mathcal{X}_h}\left(\phi^{(h)}(y, \pi_x(y))^\top w_x^{(h)} - \phi_h^\star(y, \pi_x(y))^\top w_x\right)^2 \mu_h^\star(y)^\top \phi_{h-1}^{\star,\tilde\pi}\mathrm{d}\nu(y), \quad \text{(by Lemma F.4)}$$

$$\le \alpha^{-1}\sum_{\tilde\pi\in\Psi^{(h)}}\sum_{a\in\mathcal{A}}\int_{\mathcal{X}_h}\left(\phi^{(h)}(y, a)^\top w_x^{(h)} - \phi_h^\star(y, a)^\top w_x\right)^2 \mu_h^\star(y)^\top \phi_{h-1}^{\star,\tilde\pi}\mathrm{d}\nu(y),$$

$$= A\alpha^{-1}d\cdot\mathbb{E}^{\mathsf{unif}(\Psi^{(h)})}\left[\left(\phi^{(h)}(\boldsymbol{x}_h, \boldsymbol{a}_h)^\top w_x^{(h)} - \phi_h^\star(\boldsymbol{x}_h, \boldsymbol{a}_h)^\top w_x\right)^2\right], \tag{42}$$

where the last step follows by the definition of $\Psi^{(h)}$ in Algorithm 1 and that $|\Psi^{(h)}| = d$. Now, since $w_x = \int_{\mathcal{X}_{h+1}} f(y)\mu_{h+1}^\star(y)\mathrm{d}\nu(y)$ (see (38)) and $f \in \mathcal{F}$ (see (39)); the guarantee for RepLearnin Corollary F.2 together with (42) implies that (conditioned on the event $\mathcal{E}$)

$$\left|\langle w_x^{(h)}\rangle\phi^{(h),\pi_x} - \langle w_x\rangle\phi_h^{\star,\pi_x}\right| \le \sqrt{\frac{Ad\eta^2}{64\alpha A^2 d^2}} \le \frac{\eta}{4}. \tag{43}$$

**Applying the guarantee for RobustSpanner.** Letting $\pi_1,\ldots,\pi_d$ be the policies returned by RobustSpanner at iteration $h$ of SpanRL, the guarantee of RobustSpanner in Lemma F.3 implies that there exist $\beta_1,\ldots,\beta_d \in [-2, 2]$ such that

$$\left\|\phi^{(h),\pi_x} - \sum_{i=1}^d \beta_i\phi^{(h),\pi_i}\right\| \le 3d\varepsilon \le \frac{\eta}{12d^{3/2}}, \tag{44}$$

where the last inequality follows by the fact that $\varepsilon = \frac{\eta}{36d^{5/2}}$. Combining (44) with (43) and using the triangle inequality, we get that

$$w_x^\top \phi_h^{\star,\pi_x} \le \sum_{i=1}^d \beta_i w_x^\top \phi_h^{\star,\pi_i} + \|w_x^{(h)}\|\cdot\frac{\eta}{12d^{3/2}} + \frac{\eta}{4},$$

$$\le \sum_{i=1}^d \beta_i w_x^\top \phi_h^{\star,\pi_i} + \frac{\eta}{4} + \frac{\eta}{4}, \quad \text{(by (41))}$$

$$\le 2d\max_{i\in[d]} w_x^\top \phi_h^{\star,\pi_i} + \frac{\eta}{2}.$$

Combining this with (40) and rearranging implies

$$w_x^\top \phi_h^{\star,\pi_x} \le 4d\cdot\max_{i\in[d]} w_x^\top \phi_h^{\star,\pi_i}. \tag{45}$$

On the other hand, by definition of $w_x$, we have

$$\max_{i\in[d]} w_x^\top \phi_h^{\star,\pi_i} = \max_{i\in[d]} \theta_x^\top \phi_{h+1}^{\star,\pi_i\circ_{h+1}\pi_x},$$

$$= \frac{1}{\|\mu_{h+2}^\star(x)\|}\max_{i\in[d]}\mathbb{E}^{\pi_i\circ_{h+1}\pi_x}\left[\mu_{h+2}^\star(x)^\top \phi_{h+1}^\star(\boldsymbol{x}_{h+1}, \boldsymbol{a}_{h+1})\right],$$

$$\le \frac{A}{\|\mu_{h+2}^\star(x)\|}\max_{i\in[d]}\mathbb{E}^{\pi_i\circ_{h+1}\pi_{\mathsf{unif}}}\left[\mu_{h+2}^\star(x)^\top \phi_{h+1}^\star(\boldsymbol{x}_{h+1}, \boldsymbol{a}_{h+1})\right], \quad \text{(see below)}$$

$$= \frac{A}{\|\mu_{h+2}^\star(x)\|}\max_{\pi\in\Psi^{(h+2)}}\mu_{h+2}^\star(x)^\top \phi_{h+1}^{\star,\pi}, \tag{46}$$

where the inequality follows from the non-negativity of $\mu_{h+1}^\star(\cdot)^\top \phi_{h+1}^\star(x,a)$, for all $(x,a) \in \mathcal{X}_h \times \mathcal{A}$ (due to Lemma F.4), and (46) follows from the definition of $\Psi^{(h+2)}$ in Line 13 of Algorithm 1. Combining (45) and (46) then implies that

$$\frac{1}{\|\mu_{h+2}^\star(x)\|} \mu_{h+2}^\star(x)^\top \phi_{h+1}^{\star,\pi_x} = \theta_x^\top \phi_{h+1}^{\star,\pi_x} = w_x^\top \phi_h^{\star,\pi_x} \le 4d \cdot \max_{i \in [d]} w_x^\top \phi_h^{\star,\pi_i},$$

$$\le \frac{4Ad}{\|\mu_{h+2}^\star(x)\|} \max_{\pi \in \Psi^{(h+2)}} \mu_{h+2}^\star(x)^\top \phi_{h+1}^{\star,\pi}.$$

This, together with Lemma F.4, implies that (37) holds. Since this argument holds uniformly for all $x \in \mathcal{X}_{h+2}$, this completes the proof. $\qquad\square$

### F.6   Proof of Lemma 3.1

By definition for $x \in \mathcal{X}_{h+1}$, we have $d^\pi(x) = \mathbb{E}^\pi \left[ \mu_{h+1}^\star(x)^\top \phi_h^\star(\boldsymbol{x}_h, \boldsymbol{a}_h) \right]$. Let $\pi_x$ denote the policy maximizing $d^\pi(x)$ (if no such maximizer exists, we may pass to a maximizing sequence) and let $\Psi = \{\pi_1, \dots, \pi_d\}$. Then, we have for some $\beta_1, \dots, \beta_d \in [-C, C]$,

$$d^{\pi_x}(x) = \mu_{h+1}^\star(x)^\top \left( \sum_{i=1}^d \beta_i \phi_h^{\star,\pi_i} \right) + \mu_{h+1}^\star(x)^\top \left( \phi_h^{\star,\pi_x} - \sum_{i=1}^d \beta_i \phi_h^{\star,\pi_i} \right),$$

$$\le Cd \cdot \max_{i \in [d]} \mu_{h+1}^\star(x)^\top \phi_h^{\star,\pi_i} + \varepsilon \cdot \|\mu_{h+1}^\star(x)\|, \quad \text{(Cauchy-Schwarz)}$$

$$\le Cd \cdot \max_{i \in [d]} \mu_{h+1}^\star(x)^\top \phi_h^{\star,\pi_i} + \frac{1}{2} d^{\pi_x}(x),$$

where the inequality follows by the fact that Assumption 2.1 holds with $\varepsilon \le \eta/2$. The result now follows by rearranging.

## G   Application to Reward-Based RL

In this section, we explain how the output $\Psi^{(1:H)}$ of SpanRL (Algorithm 1) can be used to optimize downstream reward functions $r_{1:H}$; our treatment is standard. Since the output of SpanRL is a policy cover, one way to optimize the sum of rewards $S_H := \sum_{h=1}^H r_h$ is by first generating trajectories using policies in $\Psi^{(1:H)}$, then applying an offline RL algorithm, e.g. Fitted Q-Iteration (FQI) [16], to optimize $S_H$. It is also possible to use PSDP with the policy cover $\Psi^{(1:H)}$ to achieve the same goal. We will showcase the latter approach since we have already stated a guarantee for PSDP.

As in Appendix D, we assume access to a function class $\mathcal{G} \subseteq \{g : \mathcal{X} \times \mathcal{A} \to \mathbb{R}\}$ that realizes the rewards $r_{1:H}$ in the following sense: for all $h \in [H]$ and all $\pi \in \Pi_{\mathsf{M}}^{h:H}$,

$$Q_h^\pi \in \mathcal{G}, \quad \text{where} \quad Q_h^\pi(x,a) := r_h(x,a) + \mathbb{E}^\pi \left[ \sum_{t=h+1}^H r_t(\boldsymbol{x}_t, \boldsymbol{a}_t) \ \middle| \ \boldsymbol{x}_h = x, \boldsymbol{a}_h = a \right].$$

Note that when the reward functions $r_{1:H}$ are linear in the feature map $\phi_h^\star$; that is, when for all $h \in [H]$ and $(x,a) \in \mathcal{X}_h \times \mathcal{A}$, $r_h(x,a) = \theta_h^\top \phi_h^\star(x,a)$ for some $\theta_h \in \mathcal{B}(1)$ (a common assumption in the context of RL in Low-Rank MDPs [32, 31, 49, 35]), then the function class

$$\mathcal{G} := \{g : (x,a) \mapsto \phi(x,a)^\top w \mid \phi \in \Phi, w \in \mathcal{B}(2H\sqrt{d})\},$$

realizes $r_{1:H}$. Note that $\mathcal{G}$ is the same function class we used for the PSDP subroutine in Algorithm 3, albeit with a larger ball for the $w$'s. For the sake of generality, we state the next result (which shows how to use a policy cover to optimize a downstream reward function) for general $r_{1:H}$ and a function class $\mathcal{G}$ that realizes $r_{1:H}$.

**Theorem G.1.** *Let $\varepsilon > 0$ and $\delta \in (0,1)$. Suppose reward functions $r_{1:H} : \mathcal{X} \times \mathcal{A} \to \mathbb{R}$, a collection of policies $\Psi^{(1:H)}$, and a parameter $n \ge 1$ satisfy the following:*

- *The function class $\mathcal{G}$ realizes the reward functions $r_{1:H}$ (in the sense of Definition D.1), and $\lim_{n\to\infty} n^{-1} \cdot \log \mathcal{N}_\mathcal{G}(1/n) = 0$, where $\mathcal{N}_\mathcal{G}(1/n)$ to denote the $\frac{1}{n}$-covering number of $\mathcal{G}$ in the supremum norm. Furthermore, we suppose that functions in $\mathcal{G}$ are uniformly bounded by $H\sqrt{d}$.*

- *For some $0 < \alpha \le 1$, for each $1 \le h \le H$, it holds that $\Psi^{(h)}$ is an $\alpha$-policy cover for layer $h$ and moreover $|\Psi^{(h)}| \le d$.*

- *The parameter $n$ is chosen such that $cdH\alpha^{-1} \cdot \varepsilon_{\mathsf{stat}}(n, \delta/H) \le \varepsilon$, where $\varepsilon_{\mathsf{stat}}(n, \delta') \coloneqq \sqrt{dH^2 n^{-1} \cdot (\log \mathcal{N}_{\mathcal{G}}(1/n) + \log(1/\delta))}$ and $c > 0$ is a large enough absolute constant.*

*Then, with probability at least $1 - \delta$, the policy $\hat{\pi} = \mathsf{PSDP}(H, r_{1:H}, \mathcal{G}, P^{(1:H)}, n)$ (where $P^{(t)} \coloneqq \mathsf{unif}(\Psi^{(t)})$, for each $t \in [h]$) coming from [Algorithm 3](), satisfies the following guarantee:*

$$\max_{\pi \in \Pi_{\mathsf{M}}} \mathbb{E}^{\pi}\left[ \sum_{h=1}^{H} r_h(\boldsymbol{x}_h, \boldsymbol{a}_h) \right] \le \mathbb{E}^{\hat{\pi}}\left[ \sum_{h=1}^{H} r_h(\boldsymbol{x}_h, \boldsymbol{a}_h) \right] + \varepsilon.$$

*Moreover, the number of episodes used by $\mathsf{PSDP}$ in this case is*

$$\widetilde{O}\left( \frac{A^2 d^5 H^5 (\log \mathcal{N}_{\mathcal{G}}(\varepsilon) + \log(1/\delta))}{\varepsilon^2} \right).$$

**Proof.** This is simply a restatement of [Theorem D.2]() with $h = H$. The number of trajectories follows by the fact that each call to PSDP requires $Hn$ trajectories. $\qquad\square$

# H   Properties of Reachability Assumption

In this section, we compare $\eta$-reachability ([Assumption 2.1]()) to different reachability assumptions used in the literature in the context of RL in Low-Rank MDPs and show that ours is the weakest among those commonly assumed. In [Appendix H.1](), we demonstrate an exponential separation between our notion of reachability and that considered with respect to the popular *latent variable model* [1, 35]. In [Appendix H.2](), we consider a number of other reachability assumptions made outside the latent variable model and show how they imply [Assumption 2.1]().

## H.1   Comparison to Latent Variable Model

In this subsection, we show that our reachability assumption is implied a reachability assumption used by [1, 35] in the latent variable/non-negative feature model, and show that our reachability assumption can hold even when the best possible latent variable embedding dimension is exponential in the dimension $d$. We begin by defining the latent variable model.

**Definition H.1** (Latent variable representation). *Givn a transition operator $T : \mathcal{X} \times \mathcal{A} \to \Delta(\mathcal{X})$, a latent variable representation consists of a countable latent space $\mathcal{Z}$ and functions $\psi : \mathcal{X} \times \mathcal{A} \to \Delta(\mathcal{Z})$ and $q : \mathcal{Z} \to \Delta(\mathcal{X})$, such that $T(\cdot \mid x, a) = \sum_{z \in \mathcal{Z}} q(\cdot \mid z) \psi(z \mid x, a)$. The latent variable dimension of $T$, denoted $d_{\mathsf{LV}}$ is the cardinality of smallest latent space $\mathcal{Z}$ for which $T$ admits a latent variable representation.*

The interpretation for the latent variable model is as follows:

1. Each $(x, a)$ pair induces a distribution $\psi(x, a) \in \Delta(\mathcal{Z})$ over $z \in \mathcal{Z}$.

2. The latent variable is sampled as $\boldsymbol{z} \sim \psi(x, a)$.

3. The next state is sampled as $\boldsymbol{x}' \sim q(\cdot \mid \boldsymbol{z})$.

Note that in discrete state spaces, all transition operators admit a trivial latent variable representation, as we may take $\psi(x, a) = T(\cdot \mid x, a)$, but the dimension of such a representation is potentially infinite. A latent variable representation certifies that there exists a factorization $T(x' \mid x, a) = \psi(x, a)^{\top} q(x')$ with embedding dimension $|\mathcal{Z}|$, and so $d_{\mathsf{LV}}$, and hence gives an upper bound on the rank of the transition operator. On the other hand, compared with the general Low-Rank factorization, the latent variable factorization additionally requires that $\psi(x, a)$ and $q(\cdot \mid z)$ are probability distributions, and thus non-negative, for all $z \in \mathcal{Z}$ and $(x, a) \in \mathcal{X} \times \mathcal{A}$, implying that $d_{\mathsf{LV}}$ is equivalent to the *non-negative rank* [1] of the transition operator.

Assuming that a latent variable representation exists, [1, 35] consider the following notion of reachability.

**Definition H.2** (Reachability in latent variable model). *There exists $\eta > 0$ such that*

$$\forall h \in [H - 1], \forall z \in \mathcal{Z}_{h+1}, \quad \sup_{\pi \in \Pi_{\mathsf{M}}} \mathbb{P}^{\pi}[\boldsymbol{z}_{h+1} = z] \ge \eta. \tag{47}$$

We first show the latent variable reachability condition above implies our more general assumption.

**Lemma H.1.** *Consider a Low-Rank MDP $\mathcal{M}$ with rank $d \geq 1$. Under the latent variable model in Definition H.1, if the latent variable reachability condition in (47) is satisfied for some $\eta > 0$, then, for all $h \in [H]$, the transition kernel $T_h$ in $\mathcal{M}$ admits a factorization $T_h(\cdot \mid x,a) = \mu_{h+1}^{\star}(\cdot)^{\top}\phi_h^{\star}(x,a)$, where $\mu_{h+1}^{\star}(\cdot) \in \mathbb{R}^{d_{\mathsf{LV}}}$ and $\phi_h^{\star}(\cdot,\cdot) \in \mathbb{R}^{d_{\mathsf{LV}}}$, such that $d_{\mathsf{LV}} \leq dA^2/\eta^2$ and $\frac{\eta^2}{A\sqrt{d}}$-reachability (in the sense of Assumption 2.1) is satisfied.*

**Proof of Lemma H.1.** Suppose that Assumption 2.1 ($\eta$-reachability) holds. By Agarwal et al. [1, Proposition 4], the non-negative rank of $\mathcal{M}$ is bounded as $d_{\mathsf{LV}} \leq dA^2/\eta^2$.

Letting $q$ and $\psi$ be as in the definition of the latent variable representation in Definition H.1, we define $\mu_{h+1}^{\star}$ and $\phi_h^{\star}$ as: for all $h \in [H-1]$,

$$\mu_{h+1}^{\star}(\cdot) \coloneqq (q(\cdot \mid z))_{z \in \mathcal{Z}} \in \mathbb{R}^{d_{\mathsf{LV}}}, \quad \text{and} \quad \phi_h^{\star}(\cdot,\cdot) \coloneqq (\psi(z \mid \cdot,\cdot))_{z \in \mathcal{Z}} \in \mathbb{R}^{d_{\mathsf{LV}}}.$$

Now, fix $h \in [H-1]$ and $x \in \mathcal{X}_{h+1}$. For $z_0 \in \arg\max_{z \in \mathcal{Z}_{h+1}} q(x \mid z)$, we have

$$\begin{aligned}
\sup_{\pi \in \Pi_{\mathsf{M}}} d^{\pi}(x) = \mathbb{P}^{\pi}[\boldsymbol{x}_{h+1} = x] &= \sup_{\pi \in \Pi_{\mathsf{M}}} \sum_{z \in \mathcal{Z}_{h+1}} q(x \mid z) \cdot \mathbb{E}^{\pi}[\psi(z \mid \boldsymbol{x}_h, \boldsymbol{a}_h)], \\
&= \sup_{\pi \in \Pi_{\mathsf{M}}} q(x \mid z_0) \cdot \mathbb{E}^{\pi}[\psi(z_0 \mid \boldsymbol{x}_h, \boldsymbol{a}_h)], \\
&= \|\mu_{h+1}^{\star}(x)\|_{\infty} \cdot \sup_{\pi \in \Pi_{\mathsf{M}}} \mathbb{P}^{\pi}[\boldsymbol{z}_{h+1} = z_0], \\
&\geq \eta \cdot \|\mu_{h+1}^{\star}(x)\|_{\infty}, \quad \text{(using reachability)} \\
&\geq \frac{\eta}{\sqrt{d_{\mathsf{LV}}}} \cdot \|\mu_{h+1}^{\star}(x)\|.
\end{aligned}$$

$\square$

We now complement the result above by showing that there exists low-rank MDPs for which our notion of reachability (Assumption 2.1) is satisfied with $\eta$ polynomially small, yet the best possible latent variable embedding has dimension $d_{\mathsf{LV}} = 2^{\Omega(d)}$. This contrasts the results in [1, Proposition 2], which show that latent variable reachability implies a polynomial bound on the latent variable dimension.

**Theorem H.3.** *There exists a one-step Low-Rank-MDP of rank $d \geq 1$, where $\eta$-reachability (Assumption 2.1) is satisfied with $\eta = \frac{1}{2\sqrt{d}}$, but where the non-negative rank satisfies $d_{\mathsf{LV}} = 2^{\Omega(d)}$.*

**Proof of Theorem H.3.** Let $n \in \mathbb{N}$ and $d \coloneqq \binom{n}{2} + 1$. As shown in the proof of Agarwal et al. [1, Proposition 2], there exists a horizon-two MDP $\mathcal{M}$ with the following properties:

- The state spaces $\mathcal{X}_1$ and $\mathcal{X}_2$ at layers 1 and 2, respectively, are finite.
- The cardinality of $\mathcal{A}$ is $d$; i.e. $\mathcal{A} = \{a_1, \ldots, a_d\}$.[11]
- The transition kernel $T_1$ admits the factorization:

$$T_1(\cdot \mid x,a) = \mu_2^{\star}(\cdot)^{\top}\phi_1^{\star}(x,a) \in \Delta(\mathcal{X}_2), \quad \forall (x,a) \in \mathcal{X}_1 \times \mathcal{A},$$

  where for all $x' \in \mathcal{X}_2$, $\mu_2^{\star}(x') \in \mathbb{R}_{\geq 0}^d$, and for all $(x,a) \in \mathcal{X}_1 \times \mathcal{A}$, $\phi_1^{\star}(x,a) \in \mathbb{R}_{\geq 0}^d$.

- The non-negative rank of $\mathcal{M}$ is $d_{\mathsf{LV}} = 2^{\Omega(d)}$.

We augment this MDP by adding an extra state $x_0$, and let $\overline{\mathcal{X}}_1 \coloneqq \mathcal{X}_1 \cup \{x_0\}$. We define $\overline{\phi}_1^{\star} : \overline{\mathcal{X}}_1 \times \mathcal{A} \to \mathbb{R}_{\geq 0}^d$ be the extension of $\phi_1^{\star}$ given by

$$\forall i \in [d], \quad \overline{\phi}_1^{\star}(x_0, a_i) = e_i, \quad \text{and} \quad \forall x \in \mathcal{X}_1, \quad \overline{\phi}_1^{\star}(x, a_i) = \phi_1^{\star}(x, a_i),$$

---

[11]Technically, the example in the proof of [1, Proposition 2] does not explicitly specify the number of actions. Instead, the example assigns a number of state-action pairs to vectors in $\mathbb{R}^d$, without specifying the number of actions. The number of actions in their example is a degree of freedom, which we set to $d$ here without loss of generality.

where $e_i$ is the $i$th basis element in $\mathbb{R}^d$. We define the initial state distribution to have $\rho(x_0) = \frac{1}{2}$ and $\rho(x) = \frac{1}{2|\mathcal{X}_1|}$, for all $x \in \mathcal{X}_1$.[12] We let $\overline{\mathcal{M}} = (\mathcal{X}_1 \cup \mathcal{X}_2, \mathcal{A}, \overline{\phi}_1^\star, (\mu_h^\star)_{h \in [2]}, \rho)$ denote the resulting MDP. Note that adding an extra state at layer 1 in this fashion only adds $d$ additional rows to the transition matrix $T$ (viewed as a $(|\mathcal{X}_1 \times \mathcal{A}|) \times |\mathcal{X}_2|$ matrix). Therefore, the non-negative rank of $\overline{\mathcal{M}}$ is as least that of $\mathcal{M}$.

We now show that reachability is satisfied in $\overline{\mathcal{M}}$. Let $\pi_i$ the policy that always plays action $a_i$. With this, we have that for any $x' \in \mathcal{X}_2$,

$$
\begin{aligned}
\sup_{\pi \in \Pi_{\mathsf{M}}} d^\pi(x') &\geq \max_{i \in [d]} d^{\pi_i}(x'), \\
&= \max_{i \in [d]} \mu_2^\star(x')^\top \mathbb{E}[\overline{\phi}_1^\star(\boldsymbol{x}_1, a_i)], \\
&= \max_{i \in [d]} \left\{ \mathbb{E}[\mathbb{I}\{\boldsymbol{x}_1 = x_0\} \cdot \mu_2^\star(x')^\top \overline{\phi}_1^\star(\boldsymbol{x}_1, a_i)] + \mathbb{E}[\mathbb{I}\{\boldsymbol{x}_1 \neq x_0\} \cdot \mu_2^\star(x')^\top \overline{\phi}_1^\star(\boldsymbol{x}_1, a_i)] \right\}, \\
&\geq \max_{i \in [d]} \rho(x_0) \mu_2^\star(x')^\top \overline{\phi}_1^\star(x_0, a_i). \qquad (48)
\end{aligned}
$$

where the last inequality follows by the fact that, for all $(x, a) \in \mathcal{X}_1 \times \mathcal{A}$, $\mu_2^\star(\cdot)^\top \overline{\phi}_1^\star(x, a) = \mu_2^\star(x')^\top \phi_1^\star(x, a) \geq 0$ (since $\mu_2^\star(x')^\top \phi_1^\star(x, a)$ is a conditional density). On the other hand, from the construction of $\overline{\phi}_1^\star$ and the fact that $\mu_2^\star(x') \in \mathbb{R}_{\geq 0}^d$, we have

$$
\max_{i \in [d]} \mu_2^\star(x')^\top \overline{\phi}_1^\star(x_0, a_i) = \|\mu_2^\star(x')\|_\infty \geq \|\mu_2^\star(x')\| / \sqrt{d}.
$$

Combining this with (48) and using that $\rho(x_0) = 1/2$ implies that reachability $1/(2\sqrt{d})$ is satisfied in $\overline{\mathcal{M}}$. $\qquad \square$

## H.2 Relation to Other Reachability Assumptions

In this subsection, we show that Assumption 2.1 is implied by a notion of *feature coverage* used in the context of transfer learning in Low-Rank MDPs [5], as well as a notion of *explorability* used in the context of reward-free RL in linear MDPs [46].

### H.2.1 Feature Coverage

We first consider coverage condition used by Agarwal et al. [5], which involves the second moments of the feature map $\phi_h^\star$.

**Definition H.4** ($\eta$-feature coverage). *We say that the linear MDP with featurization $\phi_h^\star$ satisfies $\eta$-feature coverage if for all $h \in [H]$,*

$$
\sup_{\pi \in \Pi_{\mathsf{M}}} \lambda_{\min}\left(\mathbb{E}^\pi[\phi_h^\star(\boldsymbol{x}_h, \boldsymbol{a}_h)\phi_h^\star(\boldsymbol{x}_h, \boldsymbol{a}_h)^\top]\right) \geq \eta.
$$

We show that $\eta$-feature coverage implies $(\eta/2)^{3/2}$-reachability. Thus, up to polynomial dependence, $\eta$-feature coverage is a special case of Assumption 2.1.

**Lemma H.2.** *Suppose that an MDP satisfies $\eta$-feature coverage as in Definition H.4 for some $\eta > 0$. If $\phi_h^\star(x, a) \in \mathcal{B}(1)$ for all $x, a$, then the MDP satisfies $(\eta/2)^{3/2}$-reachability in the sense of Assumption 2.1.*

**Proof of Lemma H.2.** Let $h \in [H]$ and $x \in \mathcal{X}_{h+1}$ be given and define

$$
\theta := \frac{\mu_{h+1}^\star(x)}{\|\mu_{h+1}^\star(x)\|}.
$$

To keep notation compact, we define $\boldsymbol{\phi}_h^\star := \phi_h^\star(\boldsymbol{x}_h, \boldsymbol{a}_h)$. By $\eta$-feature coverage, there exists $\pi \in \Pi_{\mathsf{M}}$ such that

$$
\begin{aligned}
\eta \leq \mathbb{E}^\pi[(\theta^\top \boldsymbol{\phi}_h^\star)^2] &= \mathbb{E}^\pi[\mathbb{I}\{(\theta^\top \boldsymbol{\phi}_h^\star)^2 < \eta/2\} \cdot (\theta^\top \boldsymbol{\phi}_h^\star)^2] + \mathbb{E}^\pi[\mathbb{I}\{(\theta^\top \boldsymbol{\phi}_h^\star)^2 \geq \eta/2\} \cdot (\theta^\top \boldsymbol{\phi}_h^\star)^2], \\
&\leq \eta/2 + \mathbb{P}^\pi[(\theta^\top \boldsymbol{\phi}_h^\star)^2 \geq \eta/2], \qquad (49)
\end{aligned}
$$

---

[12]We note that [1] did not specify the initial distribution, which is not needed for the conclusion of their result.

where we have used that $\|\theta\| = 1$ and $\|\phi_h^\star(x, a)\| \leq 1$ for all $(x, a) \in \mathcal{X}_h \times \mathcal{A}$. Rearranging (49) and using that $\theta^\top \phi_h^\star \geq 0$ (it is a scaled conditional density), have

$$\mathbb{P}^\pi[\theta^\top \phi_h^\star \geq \sqrt{\eta/2}] = \mathbb{P}^\pi[(\theta^\top \phi_h^\star)^2 \geq \eta/2] \geq \eta/2.$$

Now, by Markov's inequality, we have that

$$\theta^\top \phi_h^{\star, \pi} = \mathbb{E}^\pi[\theta^\top \phi_h^\star] \geq \sqrt{\eta/2} \cdot \mathbb{P}^\pi[\theta^\top \phi_h^\star \geq \sqrt{\eta/2}] \geq (\eta/2)^{3/2},$$

where we have once more used that $\theta^\top \phi_h^\star \geq 0$ almost surely. $\qquad\square$

### H.2.2 Explorability

We now consider the *explorability* assumption of [46], which involves the first moment of the feature map $\phi_h^\star$. This notion is defined as follows.

**Definition H.5** ($\eta$-explorability). *We say that a linear MDP satisfies $\eta$-explorability if for any $h \in [H]$ and any $\theta \in \mathbb{R}^d \smallsetminus \{0\}$ it holds that*

$$\sup_{\pi \in \Pi_{\mathsf{M}}} |\theta^\top \mathbb{E}^\pi[\phi_h^\star(\boldsymbol{x}_h, \boldsymbol{a}_h)]| \geq \eta \cdot \|\theta\|.$$

We now show that $\eta$-explorability is a special case of $\eta$-reachability:

**Lemma H.3.** *Suppose that the explorability condition in Definition H.5 is satisfied with $\eta > 0$. Then, $\eta$-reachability is satisfied.*

**Proof of Lemma H.3.** Let $x \in \mathcal{X}_{h+1}$ and define $\theta := \mu_{h+1}^\star(x)$. By explorability, we have that

$$\begin{aligned}
\sup_{\pi \in \Pi_{\mathsf{M}}} d^\pi(x) &= \sup_{\pi \in \Pi_{\mathsf{M}}} \mathbb{E}^\pi[\mu_{h+1}^\star(x)^\top \phi_h^\star(\boldsymbol{x}_h, \boldsymbol{a}_h)], \\
&= \sup_{\pi \in \Pi_{\mathsf{M}}} |\mathbb{E}^\pi[\mu_{h+1}^\star(x)^\top \phi_h^\star(\boldsymbol{x}_h, \boldsymbol{a}_h)]|, \quad (\mu_{h+1}^\star(\cdot)^\top \phi_h^\star(x, a) \text{ is a condition law}) \\
&= \sup_{\pi \in \Pi_{\mathsf{M}}} |\theta^\top \mathbb{E}^\pi[\phi_h^\star(\boldsymbol{x}_h, \boldsymbol{a}_h)]|, \\
&\geq \eta \cdot \|\theta\|, \quad \text{(by explorability)} \\
&= \eta \cdot \|\mu_{h+1}^\star(x)\|.
\end{aligned}$$

This shows that Assumption 2.1 is satisfied with parameter $\eta$. $\qquad\square$

