# OpenReview forum: "Efficient Model-Free Exploration in Low-Rank MDPs"
_NeurIPS.cc/2023/Conference — NeurIPS 2023 poster_

### Official Review · Reviewer_qxYM · 2023-06-15

**Soundness:** 3 good
**Presentation:** 3 good
**Contribution:** 3 good
**Rating:** 6
**Confidence:** 3

**Summary:**

This paper studied model-free exploration in low-rank MDP. The authors proposed an algorithm based on the idea of barycentric spanner, which is provably efficient given a reachability assumption weaker than previous literatures.

**Strengths:**

The paper writing is clear. The comparsion with previous related work and the design idea of the algorithms are clear.

The paper provided results with a weaker reachability assumption than previous literatures.

**Weaknesses:**

I have doubt about whether the algorithm can be called "efficient". It still requires to solve a complex optimization problem in Eq.(6), especially the objective function can be non-convex/concave w.r.t. feature class $\Phi$ and function class $\mathcal{F}$, even though many previous literatures claimed that "it can be efficiently solved".

The sample complexity should be improved.

Although this paper considered the model-free setting and realizability of $\phi^*$ is much milder than the model realizability, it still requires reachability assumption in some sense, while the previous model-based algorithms (like [1]) does not.

[1] Uehara et. al., Representation Learning for Online and Offline RL in Low-rank MDPs

**Questions:**

See weakness.

**Limitations:**

N.A.

---

> ### Author Rebuttal · Authors · 2023-08-10
>
> Thank you for the review and useful suggestions!
>
> **“I have doubt about whether the algorithm can be called "efficient"...”**
> Regarding the computational efficiency of our algorithm, as noted, many other works use the same minimax oracle (in fact, our oracle is slightly weaker because the maximum is taken over a simpler function class). While considering alternatives to this oracle is an interesting direction for future research, several works have empirically demonstrated the viability of this minimax oracle. Note also, that even after assuming such an oracle, the problem of how to use this oracle to rigorously explore in the low-rank setting is far from trivial.
>
> **“The sample complexity should be improved.”**
> Regarding the sample complexity, with a slight modification to the current proof, we have figured out how to improve the sample complexity to $O(1/\varepsilon^2)$. We will include this in the revision.
>
> **“Although this paper considered the model-free setting… it still requires reachability assumption in some sense, while the previous model-based algorithms (like [1]) does not.”**
> While it is true that model-based algorithms do not require reachability, note that assuming access to a model class is a nontrivial extra assumption. Indeed, it is often thought that learning the model is significantly more challenging than learning the feature map $\phi_\star$. Consider the task of learning to control in robotics or atari with images (raw pixels) as observations. Here, model-based methods must learn a generative model mapping latent states to images, whereas model-free methods like SpanRL only need to learn a mapping from images to latent states; it is clear that the former task is significantly more challenging than the latter. Thus, it is of both theoretical and practical interest to develop an algorithm that does not rely on a model class in order to achieve a smaller sample complexity. Removing reachability is a natural next step, but this is somewhat orthogonal to our main contribution. We suspect this can be achieved using a more refined analysis and techniques from [Mhammedi et al. 2023] which specifically looks at removing reachability in the Block MDP setting.
>
> **References:**
>
> Zakaria Mhammedi, Dylan J Foster, and Alexander Rakhlin. “Representation learning with multi-step inverse kinematics: An efficient and optimal approach to rich-observation RL”. International Conference on Machine Learning (ICML), 2023.

---

> > ### Comment · Reviewer_qxYM · 2023-08-14
> >
> > Thanks for addressing my questions and concerns. I would like to keep my score.

---

### Official Review · Reviewer_y7X6 · 2023-07-02

**Soundness:** 3 good
**Presentation:** 3 good
**Contribution:** 3 good
**Rating:** 7
**Confidence:** 3

**Summary:**

The paper studies the reward-free exploration problem for low-rank MDPs. The approach is based on the combination of the barycentric spanner and a model-free representation module. The authors provide a theoretical analysis of the algorithm for both the reward-free exploration task and the downstream reward-based task.

**Strengths:**

The writing is clear and easy to follow. Both the representation learning problem and the reward-free exploration problem are important in RL literature. The result is significant, which applies to more general settings compared to existing works, and also has a comparable theoretical guarantee. The idea of combining barycentric spanner with representation learning is novel.

**Weaknesses:**

1. While the paper studies the exploration problem of low-rank MDPs, its contribution mainly lies in the planning phase. The paper has limited contribution on the representation learning part, which simply reuses the algorithm and theoretical guarantee from existing RL papers.

2. The main theorem provides a guarantee on the total number of episodes. It might make the result more clear if the number of samples / trajectories can also be stated, just as the theorems in previous papers. Since the algorithm repeatedly collects samples in each episode, some being hidden in the RobustSpanner module, it is not easy to figure out the sample complexity at first look.

3. The reward-based downstream task is slightly different from existing reward-free framework. A typical reward-free exploration algorithm would only collect data in the exploration phase, and only use these collected data to construct a good policy for an arbitrary reward function. In the setting of this paper, the planning phase relies on the PSPD module, which seems to need additional interaction with the environment because the module itself (Algo 3, line 4-5) needs to sample data and construct a separate dataset. It's not clear whether this additional interaction is avoidable.

**Questions:**

1. Since the representation learning part is used as a plug-in module of the algorithm, one may wonder what the theoretical guarantee will be like when the true state-action feature is given. Furthermore, how does this result compare with the existing literature for known state-action features?

2. The paper proposes a new reachability notion. While sufficient comparison has been given between it and existing notions, will this notion reduce to a more familiar concept when considering simple MDP settings, such as tabular MDP or block MDP?

**Limitations:**

The paper addressed some of its limitations and proposed some open questions for future work.

---

> ### Author Rebuttal · Authors · 2023-08-10
>
> Thank you for the positive comments and helpful suggestions!
>
> **“While the paper studies the exploration problem of low-rank MDPs, its contribution mainly lies in the planning phase…”**
> Thank you for your review. We agree that our primary algorithmic contribution is not in the representation learning aspect (although our representation learning is slightly simpler than that found in earlier work due to the simpler discriminator class that we use), but rather in providing a rigorous algorithm for *exploration* in low-rank MDPs.  We emphasize that while the representation learning objective itself is not new, our analysis *is* new, and shows for the first time that the objective can lead to meaningful guarantees in general Low-Rank MDPs without latent variable structure or Block MDP structure. We view this as an important conceptual contribution.
>
> **“The main theorem provides a guarantee on the total number of episodes…”**
> We use the term “episode” and “trajectory” interchangeably, and the sample complexity bound stated in the main theorem refers to the total number of trajectories. We will clarify this in the revision.
>
> **“The reward-based downstream task is slightly different from existing reward-free framework…”**
> Online access to the MDP for optimizing downstream rewards is not actually required. In fact, it is possible to use Fitted-Q-Iteration (FQI) [Ersnt et al. 2005] instead of PSDP for a purely offline planning phase once trajectories are generated using the policy cover produced by our algorithm. The only reason we opted to present the reward-based results using PSDP was to keep the length of the paper down (as we already use PSDP within the analysis of SpanRL). We are happy to include a proof that FQI can be used to optimize downstream rewards for a purely offline planning phase in the final revision if this is of interest to the reviewer.
>
> **“Since the representation learning part is used as a plug-in module of the algorithm…”**
> Note that if the true feature map is given, then this reduces to the linear MDP setting and our algorithm can be run in the same way, but with the representation learning step removed.  The algorithm would produce a policy cover, which could then be used within PSDP to optimize any downstream rewards. However, compared to existing linear MDP algorithms such as LSVI-UCB [Jin et al. 2019], our algorithm would be non-optimistic and would require reachability as an additional assumption. We suspect that reachability can be removed using a more refined analysis and techniques from [Mhammedi et al. 2023] which specifically deals with this.
>
>
> **“The paper proposes a new reachability notion… will this notion reduce to a more familiar concept when considering simple MDP settings, such as tabular MDP or block MDP?”**
> Essentially all existing reachability assumptions for low-rank MDPs that we are aware of reduce to the standard reachability assumption for tabular/Block MDPs [Misra et al. 2020, Mhammedi et al. 2023]. This is perhaps most easily seen through our assumption (Assumption 2.1), which we now elucidate. Observe that in the Block MDP setting, we can take $\mu_\star(x)= q(x\mid \psi_\star(x)) \cdot e_{\psi_\star(x)} \in \mathbb{R}^S$ [Du et al. 2021], where $\psi_\star$ denotes the true decoder in the Block MDP setup and $q(x | \psi_\star(x))$ is the true emission distribution. Using this choice of $\mu_\star$, along with the fact that $d^{\pi}(x)= q(x\mid \psi_\star(x)) \cdot d^{\pi}(\psi_\star(x))$, one can see that our reachability assumption, which asserts that $\max_{\pi \in \Pi} d^{\pi}(x) \geq \eta \cdot \lVert\mu_{\star}(x)\rVert$, for all $x$ can equivalently be rewritten as $\max_{\pi \in \Pi} d^{\pi}(\psi_\star(x)) \geq \eta$ for all $x$, after cancelling $q(x\mid \psi_\star(x))$ and using that $\lVert e_{\psi_\star}(x)\rVert=1$. This is equivalent to the reachability assumption in the Block MDP setting [Misra et al. 2020, Mhammedi et al. 2023].
>
>
> **References:**
>
> Damien Ernst, Pierre Geurts, and Louis Wehenkel. "Tree-based batch mode reinforcement learning." Journal of Machine Learning Research 6 (2005).
>
> Dipendra Misra, Mikael Henaff, Akshay Krishnamurthy, and John Langford. "Kinematic state abstraction and provably efficient rich-observation reinforcement learning." In International conference on machine learning, pp. 6961-6971. PMLR, 2020.
>
> Chi Jin, Zhuoran Yang, Zhaoran Wang, and Michael I. Jordan. "Provably efficient reinforcement learning with linear function approximation." In Conference on Learning Theory, pp. 2137-2143. PMLR, 2020.
>
> Zakaria Mhammedi, Dylan J Foster, and Alexander Rakhlin. “Representation learning with multi-step inverse kinematics: An efficient and optimal approach to rich-observation RL”. International Conference on Machine Learning (ICML), 2023.

---

> > ### Comment · Reviewer_y7X6 · 2023-08-14
> >
> > I would like to thank the authors for the response. My concerns have been sufficiently addressed. My score will remain the same.

---

### Official Review · Reviewer_4TUV · 2023-07-05

**Soundness:** 3 good
**Presentation:** 3 good
**Contribution:** 3 good
**Rating:** 6
**Confidence:** 3

**Summary:**

This paper studies model-free RL under low-rank MDPs. This paper first generalizes previous fromulation of low-rank MDPs to a uncountable state case. Then this paper proposes a non-optimistic, model-free RL algorithm called SpanRL, which can provide a good policy cover with polynomial sample complexity. With policy cover produced by SpanRL,  this paper optimize any downstream reward function with polynomial sample complexity.

**Strengths:**

1. This paper provides a comprehensive summary of prior research on low-rank MDPs and presents a clear table that compares their sample complexity results with previous work.

2. This paper introduces a novel model-free exploration algorithm for general low-rank MDPs, which improves previous work on model-free RL under low-rank MDPs in two aspects: requiring a less stringent reachability assumption and removing the non-negativity requirement for feature embeddings.


**Weaknesses:**

1. One advantage of the algorithm emphasized in this paper is its computational efficiency. However, there is no experiments to validate this claim.

2. Due to the page limitations, several algorithms (subroutines) are deferred to the Appendix, which makes it hard to track the algorithmic steps. Additionally, I find there is no detailed descriptions and discussions of these algorithms. A high-level description of the algorithms will be helpful for reader to understand the novelty these algorithms. For instance, algorithm 2 is an error-tolerant variant of the classical spanner computation algorithm of [1], it would be beneficial if the authors can discuss in high level about the design modification and novelty of algorithm 2.

3. The machinism why SpanRL can improve previous work is not explain well. For example, which design of algorithm helps to remove the non-nagetivity assumption in [2]? In other words, why does [2] need such assumption while SpanRL doesn't need such an assumption?

[1] B. Awerbuch and R. Kleinberg. Online linear optimization and adaptive routing. Journal of Computer and System Sciences, 74(1):97–114, 2008.

[2] A. Modi, J. Chen, A. Krishnamurthy, N. Jiang, and A. Agarwal. Model-free representation
learning and exploration in low-rank mdps. CoRR, abs/2102.07035, 2021.

**Questions:**

As listed above.

**Limitations:**

There is no negative impact in this work.

---

> ### Author Rebuttal · Authors · 2023-08-10
>
> Thank you for the positive comments and helpful suggestions!
>
> **“One advantage of the algorithm emphasized in this paper is its computational efficiency. However, there is no experiments to validate this claim.”**
> Regarding the lack of experiments, the contribution of the paper is primarily theoretical, and we believe that our theoretical contributions alone are significant enough to merit acceptance. Note however that because recent papers demonstrate the empirical success of BRIEE [Zhang et al. 2022] and MusIK [Mhammedi et al. 2023] and, because our computational oracles are identical or simpler than those found in these works, we believe there are strong reasons to expect that the assumptions underlying our work hold, and that the relevant oracles can be implemented in a straightforward and practically effective way.
>
>
> **“Due to the page limitations, several algorithms (subroutines) are deferred to the Appendix…”**
> Thank you for the suggestion. In the revision, we will include a more detailed discussion of the motivation behind our algorithm design in the main body (especially Algorithm 2, which was deferred to the appendix due to space). Note that Algorithm 2—the robust variant of the barycentric spanner algorithm—is meaningfully different from the original algorithm of Awerbuch and Kleinberg. Most notably, the algorithm “fattens” the set in directions that would otherwise be too “skinny” for one to apply an approximate minimization oracle (as opposed to an exact one); note that without this fattening step ( i.e. naively applying the algorithm from Awerbuch and Kleinberg with an approximate minimization oracle), the algorithm would not work due to the presence of these “skinny” directions, necessitating our novel variant.
>
>
> **“The machinism why SpanRL can improve previous work is not explain well…?”**
> Prior works that make use of the same representation learning objective (BRIEE [Zhang et al. 2022] and MOFFLE [Modi et al. 2021]) do not make use of spanners; instead, they appeal to exploration strategies based on elliptic bonuses, addressing the issue of approximation errors through additional assumptions (non-negativity of the factorization for MOFFLE, and Block MDP structure for BRIEE). Perhaps the most important observation in our proof is that barycentric spanners are robust to the average-case approximation error guarantee in (5) as-is, without additional structural assumptions. Intuitively, this benefit seems to arise from the fact that the spanner property only concerns the first moment of the feature map, while algorithms based on elliptic bonuses require approximation guarantees for the second moment; understanding this issue more deeply is an interesting question for future work.
>
>
>
> **References:**
>
> Xuezhou Zhang, Yuda Song, Masatoshi Uehara, Mengdi Wang, Alekh Agarwal, and Wen Sun. “Efficient reinforcement learning in block mdps: A model-free representation learning approach”. In International Conference on Machine Learning, pages 26517–26547. PMLR, 2022.
>
> Aditya Modi, Jinglin Chen, Akshay Krishnamurthy, Nan Jiang, and Alekh Agarwal. “Model-free representation learning and exploration in low-rank mdps”. CoRR, abs/2102.07035, 2021.
>
> Zakaria Mhammedi, Dylan J Foster, and Alexander Rakhlin. “Representation learning with multi-step inverse kinematics: An efficient and optimal approach to rich-observationRL”l. International Conference on Machine Learning (ICML), 2023.

---

> > ### Comment · Reviewer_4TUV · 2023-08-13
> > **Thank you**
> >
> > I thank the authors for carefully responding to all my questions/concerns. Intutively, requiring guarantees for the second moment will require more assumptions, this makes sense to me. But I am curious about deeper connection between these. I also understand this may not be answered in this paper and can be an interesting further question. Hence, I increase my score appropriately.

---

### Official Review · Reviewer_VYmd · 2023-07-17

**Soundness:** 3 good
**Presentation:** 2 fair
**Contribution:** 3 good
**Rating:** 6
**Confidence:** 3

**Summary:**

The paper proposes a new algorithm for the low-rank MDPs setting. With a new reachability assumption, the paper shows that one can learn a policy cover in a statistically and computationally efficient manner. The proposed algorithm works in a modular manner, using barycentric spanner for exploration and the replearn algorithm for representation learning. The paper proves that the proposed algorithm can indeed learn a policy cover with in poly number of samples.

**Strengths:**

1. The overall intuition of the proposed algorithm is reasonable, the design of the algorithm is well motivated, and the technical parts of the paper seems right.

2. A computationally tractable model-free algorithm is a very good contribution for the low-rank MDPs setting.

3. The reachability assumption is novel and general, but it might require a better explanation for the intuition of the such reachability assumption: while previous visitation density or feature coverage assumptions are more intuitive, using $\mu$ seems less straightforward.

4. The proposed algorithm is modular and utilizes previous algorithms well.

**Weaknesses:**

1. Unlike previous papers such as BRIEE or MUSIK, the proposed computational efficient algorithm is not empirically tested, but this is a rather small shortcoming since the major contribution of the paper is theoretical.

2. The paper requires more polishing. The ending of the paper seems rather sudden and abrupt, although plenty discussions have been given across the paper. There are other minor presentation issues, such as the FLAMBE paper is cited in two different versions.

**Questions:**

One argument that consistently appears across the paper is that the merit for a model-free algorithm is to avoid making assumption on an additional function class $\Upsilon$, could you elaborate what is the practical significance of such condition?

---

> ### Author Rebuttal · Authors · 2023-08-10
>
> Thank you for the positive comments.
>
> **“Unlike previous papers such as BRIEE or MUSIK, the proposed computational efficient algorithm is not empirically tested…”**
> Regarding the lack of experiments, the contribution of the paper is primarily theoretical, and we believe that our theoretical contributions alone are significant enough to merit acceptance. Note however that because recent papers demonstrate the empirical success of BRIEE [Zhang et al. 2022] and MusIK [Mhammedi et al. 2023] and, because our computational oracles are identical or simpler than those found in these works, we believe there are strong reasons to expect that the assumptions underlying our work hold, and that the relevant oracles can be implemented in a straightforward and practically effective way.
>
> **“The paper requires more polishing. The ending of the paper seems rather sudden and abrupt, although plenty discussions have been given across the paper…”**
> Thank you for the suggestion. In the revision, we will be sure to add a conclusion and improve any minor presentation issues.
>
> **“One argument that consistently appears across the paper is that the merit for a model-free algorithm…  could you elaborate what is the practical significance of such condition?”**
> The fact that assuming access to a model class constitutes a non-trivial additional assumption is widely acknowledged in the literature (e.g., Zhang et al. 2022, Mhammedi et al. 2023). Consider the task of learning to control in robotics or atari with images (raw pixels) as observations. Here, model-based methods must learn a generative model mapping latent states to images, whereas model-free methods like SpanRL only need to learn a mapping from images to latent states; it is clear that the former task is significantly more challenging than the latter. Thus, it is of both theoretical and practical interest to develop an algorithm that does not rely on a model class in order to achieve a smaller sample complexity.
>
> **References:**
>
> Xuezhou Zhang, Yuda Song, Masatoshi Uehara, Mengdi Wang, Alekh Agarwal, and Wen Sun. “Efficient reinforcement learning in block mdps: A model-free representation learning approach”. In International Conference on Machine Learning, pages 26517–26547. PMLR, 2022.
>
> Zakaria Mhammedi, Dylan J Foster, and Alexander Rakhlin. “Representation learning with multi-step inverse kinematics: An efficient and optimal approach to rich-observation RL”. International Conference on Machine Learning (ICML), 2023.

---

### Decision · Program_Chairs · 2023-09-21

**Decision:**

Accept (poster)

**Comment:**

After discussion, all reviewers agree that the new model-free algorithm for low-rank MDP is interesting, and the paper should be accepted.